# Variational Gaussian Processes For Linear Inverse Problems

**Thibault Randrianarisoa**
Department of Decision Sciences
Bocconi University
via Roentgen 1, 20136, Milano, MI, Italy
`thibault.randrianarisoa@unibocconi.it`

**Botond Szabo**
Department of Decision Sciences
Bocconi University
via Roentgen 1, 20136, Milano, MI, Italy
`botond.szabo@unibocconi.it`

## Abstract

By now Bayesian methods are routinely used in practice for solving inverse problems. In inverse problems the parameter or signal of interest is observed only indirectly, as an image of a given map, and the observations are typically further corrupted with noise. Bayes offers a natural way to regularize these problems via the prior distribution and provides a probabilistic solution, quantifying the remaining uncertainty in the problem. However, the computational costs of standard, sampling based Bayesian approaches can be overly large in such complex models. Therefore, in practice variational Bayes is becoming increasingly popular. Nevertheless, the theoretical understanding of these methods is still relatively limited, especially in context of inverse problems. In our analysis we investigate variational Bayesian methods for Gaussian process priors to solve linear inverse problems. We consider both mildly and severely ill-posed inverse problems and work with the popular inducing variables variational Bayes approach proposed by Titsias [57]. We derive posterior contraction rates for the variational posterior in general settings and show that the minimax estimation rate can be attained by correctly tunned procedures. As specific examples we consider a collection of inverse problems including the heat equation, Volterra operator and Radon transform and inducing variable methods based on population and empirical spectral features.

## 1 Introduction

In inverse problems we only observe the object of interest (i.e. function or signal) indirectly, through a transformation with respect to some given operator. Furthermore, the data is typically corrupted with measurement error or noise. In practice the inverse problems are often ill-posed, i.e. the inverse of the operator is not continuous. Based on the level of ill-posedness we distinguish mildly and severely ill-posed cases. The ill-posedness of the problem prevents us from simply inverting the operator as it would blow up the measurement errors in the model. Therefore, to overcome this problem, regularization techniques are applied by introducing a penalty term in the maximum likelihood approximation. Standard examples include generalized Tikhonov, total variation and Moore-Penrose estimators, see for instance [4; 5; 9; 11; 18; 56] or a recent survey [3] on data-driven methods for solving inverse problems

An increasingly popular approach to introduce regularity to the model is via the Bayesian paradigm, see for instance [3; 7; 12; 29; 54] and references therein. Beside regularization, Bayesian methods provide a probabilistic solution to the problem, which can be directly used to quantify the remaining uncertainty of the approach. This is visualised by plotting credible sets, which are sets accumulating prescribed percentage of the posterior mass. For computing the posterior typically MCMC algorithms are used, however, these can scale poorly with increasing sample size due to the complex structure of the likelihood. Therefore, in practice often alternative, approximation methods are used. Variational

Bayes (VB) casts the approximation of the posterior into an optimization problem. The VB approach became increasingly popular to scale up Bayesian inverse problems, see for instance the recent papers [22; 31; 34; 42] and references therein. However, until recently these procedures were considered black box methods basically without any theoretical underpinning. Theoretical results are just starting to emerge [2; 46; 63; 64; 66], but we still have limited understanding of these procedures in complex models, like inverse problems.

In our analysis we consider Gaussian process (GP) priors for solving linear inverse problems. For Gaussian likelihoods, due to conjugacy, the corresponding posterior has an analytic form. Nevertheless, they are applied more widely, in non-conjugate settings as well. However, training and prediction even in the standard GP regression model, scales as $O\left(n^3\right)$ (or $O\left(n^2\right)$ for exact inference in the recent paper [16] leveraging advances in computing hardware) and $O\left(n^2\right)$, respectively, which practically limits GPs to a sample size $n$ of order $10^4$. Therefore, in practice often not the full posterior, but an approximation is computed. Various such approximation methods were proposed based on some sparse or low rank structure, see for instance [13; 14; 33; 43; 49; 50; 51; 52; 57]. Our focus here lies on the increasingly popular inducing variable variational Bayes method introduced in [57; 58].

In our work we extend the inducing variable method for linear inverse problems and derive theoretical guarantees for the corresponding variational approximations. More concretely we adopt a frequentist Bayes point-of-view in our analysis by assuming that there exists a true data generating functional parameter of interest and investigate how well the variational posterior can recover this object. We derive contraction rates for the VB posterior around the true function both in the mildly and severely ill-posed inverse problems. We then focus on two specific inducing variable methods based on the spectral features of the prior covariance kernel. We show that for both methods if the number of inducing variables are chosen large enough for appropriately tunned priors the corresponding variational posterior concentrates around the true function with the optimal minimax estimation rate. One, perhaps surprising aspect of the derived results is that the number of inducing variables required to attain the optimal, minimax contraction rate is sufficiently less in the inverse setting than in the direct problem. Therefore, inverse problems can be scaled up at a higher degree than standard regression models.

*Related literature.* The theory of Bayesian approaches to linear inverse problems is now well established. The study of their asymptotic properties started with the study of conjugate priors [1; 15; 20; 26; 27; 28] before addressing the non-conjugate case [25; 45] and rate-adaptive priors [26; 55]. By now we have a good understanding of both the accuracy of the procedure for recovering the true function and the reliability of the corresponding uncertainty statements. The theory of Bayesian non-linear inverse problems is less developed, but recent years have seen an increasing interest in the topic, see the monograph [36] and references therein. Some algorithmic developments for variational Gaussian approximations in non-linear inverse problems, and applications to MCMC sampling, can be found in [40; 41].

The inducing variable approach for GPs proposed by [57; 58] has been widely used in practice. Recently, their theoretical behaviour was studied in the direct, nonparametric regression setting. In [8] it was shown that the expected Kullback-Leibler divergence between the variational class and posterior tends to zero when sufficient amount of inducing variables were used. Furthermore, optimal contraction rates [37] and frequentist coverage guarantees [38; 59; 60] were derived for several inducing variable methods. Our paper focuses on extending these results to the linear inverse setting.

*Organization.* The paper is organized as follows. In Section 2 we first introduce the inverse regression model where we carry out our analysis. Then we discuss the Bayesian approach using GPs and its variational approximations in Sections 2.1 and 2.2, respectively. As our main result we derive contraction rates for general inducing variable methods, both in the mildly and severely ill-posed cases. Then in Section 2.3 we focus on two specific inducing variable methods based on spectral features and provide more explicit results for them. We apply these results for a collection of examples, including the Volterra operator, the heat equation and the Radon transform in Section 3. Finally, we demonstrate the applicability of the procedure in the numerical analysis of Section 4 and conclude the paper with discussion in Section 5. The proof of the main theorem together with technical lemmas and additional simulation study are deferred to the supplementary material.

*Notation.* Let $C, c$ be absolute constants, independent of the parameters of the problem whose values may change from line to line. For two sequences $(a_n)$ and $(b_n)$ of numbers, $a_n \lesssim b_n$ means that there

exists a universal constant $c$ such that $a_n \leq cb_n$ and we write $a_n \asymp b_n$ if both $a_n \lesssim b_n$ and $b_n \lesssim a_n$ hold simultaneously. We denote by $a_n \ll b_n$ if $|a_n/b_n|$ tends to zero. The maximum and minimum of two real numbers $a$ and $b$ are denoted by $a \vee b$ and $a \wedge b$, respectively. We use the standard notation $\delta_{ij} = \mathbb{1}_{i=j}$. For $m \geq 1$, we note $\boldsymbol{S}_{++}^m$ the set of positive-definite matrices of size $m \times m$.

## 2 Main results

In our analysis we focus on the non-parametric random design regression model where the functional parameter is observed through a linear operator. More formally, we assume to observe i.i.d. pairs of random variables $(x_i, Y_i)_{i=1,\dots,n}$ satisfying

$$Y_i = (\mathcal{A}f_0)(x_i) + Z_i, \qquad Z_i \overset{iid}{\sim} N(0, \sigma^2), \ x_i \overset{iid}{\sim} G \qquad i = 1, \dots, n, \tag{1}$$

where $f_0 \in L_2(\mathcal{T}; \mu)$, for some domain $\mathcal{T} \subset \mathbb{R}^d$ and measure $\mu$ on $\mathcal{T}$, is the underlying functional parameter of interest and $\mathcal{A}: L_2(\mathcal{T}; \mu) \mapsto L_2(\mathcal{X}; G)$, for the measure $G = \mathcal{A}\mu$ on $\mathcal{X}$, is a known, injective, continuous linear operator. In the rest of the paper we use the notation $P_{f_0}$ and $E_{f_0}$ for the joint distribution and the corresponding expectation, respectively, of the data $(X, Y) = (x_i, Y_i)_{i=1,\dots,n}$. Furthermore, we denote by $E_X, P_X, E_{Y|X}, P_{Y|X}$ the expectation/distribution under $G^{\otimes n}$ and the law of $(Y_i)_i$ given the design respectively. Finally, for simplicity we take $\sigma^2 = 1$ in our computations.

In the following, denoting $\mathcal{A}^*$ the adjoint of $\mathcal{A}$, we assume that the self-adjoint operator $\mathcal{A}^*\mathcal{A}: L_2(\mathcal{T}; \mu) \mapsto L_2(\mathcal{T}; \mu)$ possesses countably many positive eigenvalues $(\kappa_j^2)_j$ with respect to the eigenbasis $(e_j)_j$ (which is verified if $\mathcal{A}$ is a compact operator for instance). We remark that $(g_j)_j$ defined by $\mathcal{A}e_j = \kappa_j g_j$ is an orthonormal basis of $L_2(\mathcal{X}; G)$. We work on the ill-posed problem where $\kappa_j \to 0$, the rate of decay characterizing the difficulty of the inverse problem.

**Definition 1.** *We say the problem is mildly ill-posed problem of degree $p > 0$ if $\kappa_j \asymp j^{-p}$ has a polynomial decay. In the severely ill-posed problem, the rate we consider is exponential, $\kappa_j \asymp e^{-cj^p}$ for $c > 0, p \geq 1$, and $p$ is the degree of ill-posedness once again.*

In nonparametrics it is typically assumed that $f_0$ belongs to some regularity class. Here we consider the generalized Sobolev space

$$\bar{H}^\beta := \left\{ f \in L_2(\mathcal{T}; \mu) : \|f\|_\beta < \infty \right\}, \quad \|f\|_\beta^2 = \sum_j j^{2\beta} |\langle f, e_j \rangle|^2, \tag{2}$$

for some $\beta > 0$. We note that the difficulty in estimating $f_0$ from the data is twofold: one needs to deal with the observational noise, which is a statistical problem, as well as to invert the operator $\mathcal{A}$, which comes from inverse problem theory. As a result of the ill-posedness of the problem, recovering $f_0$ from the observations may suffer from problems of unidentifiability and instability. The solution to these issues is to incorporate some form of regularization in the statistical procedure. The Bayesian approach provides a natural way to incorporate regularization into the model via the prior distribution on the functional parameter. In fact penalized likelihood estimators can be viewed as the maximum a posteriori estimators with the penalty term induced by a prior. For example Tikhonov type regularizations can be related to the RKHS-norm of a Gaussian Process prior, see [35; 44] for a more detailed discussion.

### 2.1 Gaussian Process priors for linear inverse problems

We focus on the Bayesian solution of the inverse problem and exploit the Gaussian likelihood structure by considering conjugate Gaussian Process (GP) priors on $f$. A GP $\mathcal{GP}(\eta(\cdot), k(\cdot, \cdot))$ is a set of random variables $\{f(t) \mid t \in \mathcal{T}\}$, such that any finite subset follows a Gaussian distribution. The GP is described by the mean function $\eta$ and a covariance kernel $k(t, t')$. We consider centered GPs as priors (i.e. we take $\eta \equiv 0$). Then the bilinear, symmetric nonnegative-definite function $k: \mathcal{T} \times \mathcal{T} \mapsto \mathbb{R}$ determines the properties of the process (e.g., its regularity). In view of the linearity of the operator $\mathcal{A}$ the corresponding posterior distribution is also a Gaussian process. The mean and covariance function of the posterior is given by

$$\begin{aligned} t &\mapsto K_{t\mathcal{A}f} \left( K_{\mathcal{A}f\mathcal{A}f} + \sigma^2 I_n \right)^{-1} \mathbf{y}, \\ (t, s) &\mapsto k(t, s) - K_{t\mathcal{A}f} \left( K_{\mathcal{A}f\mathcal{A}f} + \sigma^2 I_n \right)^{-1} K_{\mathcal{A}fs}, \end{aligned} \tag{3}$$

where $\mathbf{y} = (y_1, \ldots, y_n)^T$, $\boldsymbol{\mathcal{A}f} = \big(\mathcal{A}f(x_i)\big)_{i=1,\ldots,n}$, $K_{\boldsymbol{\mathcal{A}f\mathcal{A}f}} = E_\Pi \boldsymbol{\mathcal{A}f\mathcal{A}f}^T \in \mathbb{R}^{n \times n}$ with $E_\Pi$ denoting the expectation with respect to the GP prior $\Pi$, $K_{t\boldsymbol{\mathcal{A}f}}^T = \big(E_\Pi \mathcal{A}f(x_i)f(t)\big)_{i=1,\ldots,n} \in \mathbb{R}^n$, see the supplement for the detailed derivation.

Due to the closed-form expressions for the posterior and the marginal likelihood, as well as the simplicity with which uncertainty quantification may be produced, GP regression has gained popularity [44]. Furthermore, the asymptotic frequentist properties of posteriors corresponding to GP priors in the direct problem, with $\mathcal{A}$ taken to be the identity operator, is well-established by now. Optimal contraction rates and confidence guarantees for Bayesian uncertainty quantification were derived in the regression setting and beyond, see for instance [10; 39; 47; 53; 48; 61; 62; 65] and references therein. In the following, we say that $\varepsilon_n$ is an $L_2$–posterior contraction rate for the posterior $\Pi[\,\cdot\,|X,Y]$ if for any $M_n \to 0$

$$E_{f_0}\Pi\left[f\colon\ \|f - f_0\|_{L_2(\mathcal{T};\mu)} \geq M_n\varepsilon_n \mid X,Y\right] \to 0.$$

In our analysis we consider covariance kernels with eigenfunctions coinciding with the eigenfunctions of the operator $\mathcal{A}^*\mathcal{A}$, i.e. we take

$$k(t,s) = \sum\nolimits_j \lambda_j e_j(t) e_j(s), \tag{4}$$

where $(\lambda_j)_j$ denote the corresponding eigenvalues. The asymptotic behaviour of the corresponding posterior has been well investigated in the literature both in the mildly and severely ill-posed inverse problems. Rate optimal contraction rates and frequentist coverage guarantees for the resulting credible sets were derived both for known and unknown regularity parameters [15; 26; 27; 28; 55]. These results were further extended for other covariance kernels where the eigenfunctions do not exactly match the eigenfunctions of the operator $\mathcal{A}$, but in principle they have to be closely related, see [1; 20; 25; 45].

However, despite the explicit, analytic form of the posterior given in (3) and the theoretical underpinning, the practical applicability of this approach is limited for large sample size $n$. The computation of the posterior involves inverting the $n$-by-$n$ matrix $K_{\boldsymbol{\mathcal{A}f\mathcal{A}f}} + \sigma^2 I_n$, which has computational complexity $O(n^3)$. Therefore, in practice often not the true posterior, but a scalable, computationally attractive approximation is applied. Our focus here is on the increasingly popular inducing variable variational Bayes method introduced in [57; 58].

## 2.2 Variational GP for linear inverse problems

In variational Bayes the approximation of the posterior is casted as an optimization problem. First a tractable class of distributions $\mathcal{Q}$ is considered, called the variational class. Then the approximation $\Psi^*$ is computed by minimizing the Kullback-Leibler divergence between the variational class and the true posterior, i.e.

$$\Psi^* = \arg\ \inf_{Q \in \mathcal{Q}} KL\left(Q\|\Pi[\,\cdot\,|X,Y]\right).$$

There is a natural trade-off between the computational complexity and the statistical accuracy of the resulting approximation. Smaller variational class results in faster methods and easier interpretation, while more enriched classes preserve more information about the posterior ensuring better approximations.

In context of the Gaussian process regression model (with the operator $\mathcal{A}$ taken to be the identity), [58] proposed a low-rank approximation approach based on inducing variables. The idea is to compress the information encoded in the observations of size $n$ into $m$ so called inducing variables. We extend this idea for linear inverse problems. Let us consider real valued random variables $\mathbf{u} = (u_1, \ldots, u_m) \in L_2(\Pi)$, expressed as measurable linear functionals of $f$ and whose prior distribution is $\Pi_u$. In view of the linearity of $\boldsymbol{u}$, the joint distribution of $(f, \mathbf{u})$ is a Gaussian process, hence the conditional distribution $f|\mathbf{u}$ denoted by $\Pi(\cdot|\mathbf{u})$, is also a Gaussian process with mean function and covariance kernel given by

$$t \mapsto K_{t\boldsymbol{u}}K_{\boldsymbol{uu}}^{-1}\boldsymbol{u} \quad \text{and} \quad (t,s) \mapsto k(t,s) - K_{t\boldsymbol{u}}K_{\boldsymbol{uu}}^{-1}K_{\boldsymbol{us}},$$

respectively, where $K_{t\boldsymbol{u}} = E_\Pi(f(t)\boldsymbol{u}) \in \mathbb{R}^m$ and $K_{\boldsymbol{uu}} = E_\Pi(\boldsymbol{uu}^T) \in \mathbb{R}^{m \times m}$. Then the posterior is approximated via a probability measure $\boldsymbol{\Psi}_u$ on $(\mathbb{R}^m, \mathcal{B}(\mathbb{R}^m))$ by $\Psi = \int \Pi[\cdot|\mathbf{u}]d\Psi_u(\mathbf{u})$, which is

absolutely continuous against $\Pi$ and satisfies $\frac{d\Psi}{d\Pi}(f) = \frac{d\Psi_u}{d\Pi_u}(\mathbf{u}(f))$. We note that the variables $\mathbf{u}$ were first considered to be point evaluations of the GP prior process before these ideas were extended to interdomain inducing variables, e.g. integral forms of the process [30; 58].

Taking $\Psi_{\boldsymbol{u}} = \mathcal{N}(\boldsymbol{\mu}_u, \Sigma)$ as a multivariate Gaussian, the corresponding $\Psi \propto \Pi(\cdot|\boldsymbol{u})\Psi_{\boldsymbol{u}}$ is a Gaussian process, with mean and covariance functions

$$t \mapsto K_{t\boldsymbol{u}}K_{\boldsymbol{u}\boldsymbol{u}}^{-1}\boldsymbol{\mu}_u, \quad \text{and} \quad (t,s) \mapsto k(t,s) - K_{t\boldsymbol{u}}K_{\boldsymbol{u}\boldsymbol{u}}^{-1}(K_{\boldsymbol{u}\boldsymbol{u}} - \Sigma)K_{\boldsymbol{u}\boldsymbol{u}}^{-1}K_{s\boldsymbol{u}}^T. \tag{5}$$

Letting $\boldsymbol{\mu}_u$ and $\Sigma$ be the free variational parameters, the variational family is taken as

$$\mathcal{Q} := \left\{ \Psi \mid \Psi_u = \mathcal{N}(\boldsymbol{\mu}_u, \Sigma_u), \ \boldsymbol{\mu}_u \in \mathbb{R}^m, \ \Sigma_u \in \boldsymbol{S}_{++}^m \right\},$$

consisting of "$m$–sparse" Gaussian processes.

By similar computations as those from [37], it can be shown that $\Pi[\cdot|X,Y]$ is equivalent to any element of $\mathcal{Q}$ (they are mutually dominated) so that the $KL$ divergence is always finite and there exists a $\Psi_u^*$, corresponding to the minimizer $\Psi^*$ of KL $(\Psi\|\Pi[\cdot|X,Y])$. Furthermore, we have

$$\frac{d\Psi^*}{d\Pi}(f) = \frac{d\Psi_{\boldsymbol{u}}^*}{d\Pi_{\boldsymbol{u}}}(\mathbf{u}) \propto exp\Big( - \frac{1}{2\sigma^2} \int \sum_{i=1}^n (Y_i - \mathcal{A}f(x_i))^2 d\Pi(f|\mathbf{u})\Big)$$

$$\propto exp\Big( - \frac{1}{2\sigma^2} \sum_{i=1}^n (Y_i - K_{\mathcal{A}f(x_i)\boldsymbol{u}}K_{\boldsymbol{u}\boldsymbol{u}}^{-1}\mathbf{u})^2\Big)$$

where $K_{\mathcal{A}f(x_i)\boldsymbol{u}} = E_\Pi \mathcal{A}f(x_i)\boldsymbol{u}^T$. One can observe that the parameters of the variational approximations are

$$\begin{aligned}
\boldsymbol{\mu}_u^* &= \sigma^{-2}K_{\boldsymbol{u}\boldsymbol{u}}\left(K_{\boldsymbol{u}\boldsymbol{u}} + \sigma^{-2}K_{\boldsymbol{u}\mathcal{A}f}K_{\mathcal{A}f\boldsymbol{u}}\right)^{-1}K_{\boldsymbol{u}\mathcal{A}f}\mathbf{y}, \\
\Sigma_u^* &= K_{\boldsymbol{u}\boldsymbol{u}}\left(K_{\boldsymbol{u}\boldsymbol{u}} + \sigma^{-2}K_{\boldsymbol{u}\mathcal{A}f}K_{\boldsymbol{u}\mathcal{A}f}\right)^{-1}K_{\boldsymbol{u}\boldsymbol{u}},
\end{aligned} \tag{6}$$

for $K_{\boldsymbol{u}\mathcal{A}f} = K_{\mathcal{A}f\boldsymbol{u}}^T = E_\Pi \mathbf{u}(\mathcal{A}\mathbf{f})^T$ the $m \times n$ matrix whose $j$th column is $K_{\mathcal{A}f(x_i)\boldsymbol{u}}$ and $(\mathcal{A}\mathbf{f})^T = (\mathcal{A}f(x_1), \ldots, \mathcal{A}f(x_n))$. Then the explicit form for the variational posterior $\Psi^*$ can be attained by plugging in the parameters (6) into the variational mean and covariance function (5). We also define $Q_{\mathcal{A}f\mathcal{A}f} = K_{\boldsymbol{u}\mathcal{A}f}^T K_{\boldsymbol{u}\boldsymbol{u}}K_{\boldsymbol{u}\mathcal{A}f}$.

We investigate the statistical inference properties of the above variational posterior distribution $\Psi^*$. More concretely we focus on how well the variational approximation can recover the underlying true functional parameter $f_0$ of interest in the indirect, linear inverse problem (1). We derive contraction rate for $\Psi^*$ both in the mildly and severely ill-posed inverse problem case. Furthermore, we consider both the standard exponential and polynomial spectral structures for the prior, i.e. we assume that the eigenvalues of the prior covariance kernel satisfies either $\lambda_j \asymp j^{-\alpha}e^{-\xi j^p}$ or $\lambda_j \asymp j^{-1-2\alpha}$ for some $\alpha \geq 0, \xi > 0$. Finally, in view of [37], we introduce additional assumptions on the covariance kernel of the conditional distribution of $f|\boldsymbol{u}$ ensuring that the variational posterior is not too far from the true posterior in Kullback-Leibler divergence.

**Theorem 1.** *Let's assume that $f_0 \in \bar{H}^\beta$ and $\|f_j\|_\infty \lesssim j^\gamma$ for $\beta > 0, \gamma \geq 0$.*

1. *In the mildly-ill posed problem where $\kappa_j \asymp j^{-p}$, $p > 0$, if $\lambda_j \asymp j^{-1-2\alpha}$ for $\alpha > 0$ and $(\alpha \wedge \beta) + p > 3/2 + 2\gamma$, the posterior contracts at the rate $\varepsilon_n^{inv} = n^{-\frac{\alpha \wedge \beta}{1+2\alpha+2p}}$.*

2. *In the severely ill-posed problem where $\kappa_j \asymp e^{-cj^p}$, $c > 0, p \geq 1$, if $\lambda_j \asymp j^{-\alpha}e^{-\xi j^p}$ for $\alpha \geq 0, \xi > 0$, the posterior contracts at the rate $\varepsilon_n^{inv} = \log^{-\beta/p} n$.*

*Furthermore, if there exists a constant $C$ independent of $n$ such that*

$$E_X \|K_{\mathcal{A}f\mathcal{A}f} - Q_{\mathcal{A}f\mathcal{A}f}\| \leq C, \quad \text{and} \quad E_X Tr\left(K_{\mathcal{A}f\mathcal{A}f} - Q_{\mathcal{A}f\mathcal{A}f}\right) \leq Cn\varepsilon_n^2, \tag{7}$$

*where $\varepsilon_n = n^{-\frac{\alpha \wedge \beta+p}{1+2\alpha+2p}}$ in 1., and $\varepsilon_n = n^{-c/(\xi+2c)}\log^{-\beta/p+c\alpha/(\xi+2c)}(n)$ in 2., $\Psi^*$ contracts around $f_0$ at the rate $\varepsilon_n^{inv}$ for the mildly and severely ill-posed problems i.e.*

$$E_{f_0}\Psi^*\left[ f \colon \|f - f_0\|_{L_2(\mathcal{T};\mu)} \geq M_n\varepsilon_n^{inv} \right] \to 0, \quad M_n \to \infty.$$

*Proof.* We provide the sketch of the proof here, the detailed derivation of the theorem is deferred to the supplementary material. In a first step, we derive posterior contraction rates around $\mathcal{A}f_0$ in empirical $L_2$-norm under fixed design. In particular, we obtain an exponential decay of the probability expectation in the form

$$E_{Y\,|\,X}\Pi\Big[f\colon\ n^{-1}\sum\nolimits_{i=1}^{n}\big(\mathcal{A}f-\mathcal{A}f_0\big)^2(x_i)\ge M_n\varepsilon_n^2\mid X,Y\Big]\mathbb{1}_{A_n}\le Ce^{-cM_n^2 n\varepsilon_n^2}, \qquad (8)$$

for arbitrary $M_n\to\infty$, where $\varepsilon_n=n^{-\frac{\alpha\wedge\beta+p}{1+2\alpha+2p}}$ in the mildly and $\varepsilon_n=n^{-\frac{c}{\xi+2c}}\log^{-\frac{\beta}{p}+\frac{c\alpha}{\xi+2c}}n$ in the severely ill-posed problems and $A_n$ is an event on the sample space $\mathbb{R}^n$ with probability tending to one asymptotically. In the mildly ill-posed case, this follows from results in [17; 62], while additional care is needed in the severely ill-posed case. As a second step, we go back to the random design setting. We show, using concentration inequalities and controlling the tail probability of GPs in the spectral decomposition, that the empirical and population $L_2$-norms are equivalent on a large enough event. This implies contraction rate with respect to the $\|\cdot\|_{L^2(\mathcal{X},G)}$-norm around $\mathcal{A}f_0$, similarly to (8). In the third step, using the previous result on the forward map, we derive contraction rates around $f_0$. To achieve this we apply the modulus of continuity techniques introduced in [25]. Notably, we extend their ideas to infinite Gaussian series priors in the severely ill-posed case as well. Since in all these steps we can preserve the exponential upper bound for the posterior contraction (on a large enough event), we can apply Theorem 5 of [46], resulting in contraction rates for the VB procedure. It requires a control of the expected KL divergence between these two distributions, which follows from our assumptions on the expected trace and spectral norm of the covariance matrix of $\boldsymbol{\mathcal{A}f}|u$, see Lemma 3 in [37] for the identity operator $\mathcal{A}$.

$\square$

We briefly discuss the above results. First of all, the $L_2(\mathcal{T};\mu)$-contraction rate of the true posterior, to the best of our knowledge, wasn't derived explicitly in the literature before, hence it is of interest in its own right. Nevertheless, the main message is that the variational posterior achieves the same contraction rate as the true posterior under the assumption (7). Note that in the mildly ill-posed inverse problem case for eigenvalues $\lambda_j\asymp j^{-1-2\beta}$ (i.e. taking $\alpha=\beta$), the posterior contracts with the minimax rate $n^{-\beta/(1+2\beta+2p)}$. Note that the $d$–dimensional case directly follows from this result when one defines the regularity class (2) and ill-posedness (Definition 1) with $\beta/d$ and $p/d$ which would imply the rate $n^{-\beta/(d+2\beta+2p)}$. Similarly in the severely-ill posed case one can achieve the minimax logarithmic contraction rate. Furthermore, the choice of the eigenvalue structure in the theorem was done for computational convenience, the results can be generalised for other choices of $\lambda_j$ as well. Though we considered the random variables $u$ fixed as we do not optimize them above, they could conceivably be considered as free variational parameters and selected at the same time as $\boldsymbol{\mu}_u$ and $\Sigma_u$.

In the next subsection we consider two specific choices of the inducing variables, i.e. the population spectral feature method and its empirical counter part. We show that under sufficient condition on the number of inducing variables condition (7) is satisfied implying the contraction rate results derived in the preceding theorem.

## 2.3 Population and empirical spectral features methods

We focus here on two inducing variables methods, based on the spectral features (i.e. eigenspectrum) of the empirical covariance matrix $K_{\boldsymbol{\mathcal{A}f\mathcal{A}f}}=E_\Pi\mathcal{A}\mathbf{f}\mathcal{A}\mathbf{f}^T$ and the corresponding population level covariance operator $(x,y)\mapsto E_\Pi\mathcal{A}f(x)\mathcal{A}f(y)$.

We start with the former method and consider inducing variables of the form

$$u_j=\sum\nolimits_{i=1}^{n}v_j^i\mathcal{A}f(x_i),\quad j=1,\ldots,m, \qquad (9)$$

where $\mathbf{v}_j=(v_j^1,\ldots,v_j^n)$ is the eigenvector of $K_{\boldsymbol{\mathcal{A}f\mathcal{A}f}}$ corresponding to the $j$th largest eigenvalue $\rho_j$ of this matrix. Similarly to the direct problem studied in [8; 37], this results in $(K_{\boldsymbol{\mathcal{A}f\mathcal{A}f}})_{ij}=\rho_j\delta_{ij}$, $(K_{\boldsymbol{\mathcal{A}fu}})_{ij}=\rho_j v_j^i$, $Q_{\boldsymbol{\mathcal{A}f\mathcal{A}f}}=\sum_{j=1}^{m}\rho_j\mathbf{v}_j\mathbf{v}_j^T$, and $K_{\boldsymbol{\mathcal{A}f\mathcal{A}f}}-Q_{\boldsymbol{\mathcal{A}f\mathcal{A}f}}=\sum_{j=m+1}^{n}\rho_j\mathbf{v}_j\mathbf{v}_j^T$. The computational complexity of deriving the first $m$ eigenvectors of $K_{\boldsymbol{\mathcal{A}f\mathcal{A}f}}$ is $\mathcal{O}(n^2m)$. This is still quadratic in $n$, which sets limitations to its practical applicability, but it can be computed for arbitrary

choices of the prior covariance operator and map $\mathcal{A}$. We also note that this choice gives the optimal rank–$m$ approximation $Q_{\mathcal{A}f\mathcal{A}f}$ of $K_{\mathcal{A}f\mathcal{A}f}$ and it was noted in [8] that it gives the minimiser of the trace and norm terms in (7).

The second inducing variables method is based on the eigendecomposition of covariance kernel $(x, y) \mapsto E_\Pi \mathcal{A}f(x)\mathcal{A}f(y)$. Let us consider the variables

$$u_j = \int_{\mathcal{X}} \mathcal{A}f(x)e_j(x)dG(x), \quad j = 1, \ldots, m. \tag{10}$$

Again, by extending the results derived in the direct problem [8] to the inverse setting, this results in $(K_{\mathcal{A}f\mathcal{A}f})_{ij} = \lambda_j \kappa_j \delta_{ij}$, $(K_{\mathcal{A}fu})_{ij} = \lambda_j \kappa_j \phi_j^i$, $Q_{\mathcal{A}f\mathcal{A}f} = \sum_{j=1}^m \lambda_j \kappa_j \phi_j \phi_j^T$, and $K_{\mathcal{A}f\mathcal{A}f} - Q_{\mathcal{A}f\mathcal{A}f} = \sum_{j=m+1}^n \lambda_j \kappa_j \phi_j \phi_j^T$, where $\phi_j = (\phi_j(x_1), \ldots, \phi_j(x_n))^T$. The computational complexity of this method is $O(nm^2)$, which is substantially faster than its empirical counter part. However, it requires the exact knowledge of the eigenfunctions of the prior covariance kernel, and therefore in general has limited practical applicability.

**Corollary 1.** *Let's assume that $f_0 \in \bar{H}^\beta$, $\|g_j\|_\infty \lesssim j^\gamma$ for $\beta > 0, \gamma \geq 0$ and in the*

1. *mildly-ill posed case $\kappa_j \asymp j^{-p}$: take prior eigenvalues $\lambda_j \asymp j^{-1-2\alpha}$ for some $\alpha > 0$, $(\alpha \wedge \beta) + p > 3/2 + 2\gamma$, number of inducing variables $m_n \geq n^{\frac{1}{(1+2p+2\alpha)}}$ and denote by $\varepsilon_n^{inv} = n^{-\frac{\alpha \wedge \beta}{1+2\alpha+2p}}$.*

2. *severely ill-posed case $\kappa_j \asymp e^{-cj^p}$: take prior eigenvalues $\lambda_j \asymp j^{-\alpha}e^{-\xi j^p}$ for $\alpha \geq 0$, $\xi > 0$, number of inducing variables $m_n^p \geq (\xi + 2c)^{-1}\log n$, and introduce the notation $\varepsilon_n^{inv} = \log^{-\beta/p} n$.*

*Then both for the population (if $\gamma = 0$ in 1.) and empirical spectral features variational methods the corresponding variational posterior distribution contracts around the truth with the rate $\varepsilon_n^{inv}$, i.e.*

$$E_{f_0}\Psi^* \left[ f: \ \|f - f_0\|_{L_2(T;\mu)} \geq M_n \varepsilon_n^{inv} \right] \to 0, \quad M_n \to \infty.$$

**Remark 1.** *In the mildly ill-posed inverse problem taking $\alpha = \beta$ results in the minimax contraction rate for $m_n \geq n^{\frac{1}{1+2p+2\alpha}}$. Note that it is substantially less compared to the direct problem with $p = 0$, hence the computation is even faster in the inverse problem case.*

## 3 Examples

In this section we provide three specific linear inverse problems as examples. The Volterra (integral) operator and the Radon transformations are mildly ill-posed, while the heat-equation is a severely ill-posed inverse problem. We show that in all cases by optimally tunning the GP prior and including enough inducing variables, the variational approximation of the posterior provides (from a minimax perspective) optimal recovery of the underlying signal $f_0$.

### 3.1 Volterra operator

First, let us consider the Volterra operator, $\mathcal{A}: L_2[0, 1] \longrightarrow L_2[0, 1]$ satisfying that

$$\mathcal{A}f(x) = \int_0^x f(s)ds, \quad \mathcal{A}^*f(x) = \int_x^1 f(s)ds. \tag{11}$$

The eigenvalues of $\mathcal{A}^*\mathcal{A}$ and the corresponding eigenbases are given by $\kappa_j^2 = (j - 1/2)^{-2}\pi^{-2}$, $e_j(x) = \sqrt{2}\cos((j - 1/2)\pi x)$, $g_j(x) = \sqrt{2}\sin((j - 1/2)\pi x)$ respectively, see [21]. Therefore the problem is mildly ill-posed with degree $p = 1$ and these bases are uniformly bounded, i.e. $\sup_j \|e_j\|_\infty \vee \|g_j\|_\infty < \infty$. The following lemma is then a direct application of Corollary 1.

**Corollary 2.** *Consider the Volterra operator in (1) and assume that $f_0 \in \bar{H}^\beta$, for some $\beta > 1/2$. Set the eigenvalues in (4) as $\lambda_j = j^{-1-2\beta}$. Then the variational posterior $\Psi^*$ resulting from either the empirical or population spectral features inducing variable methods achieves the minimax contraction rate if the number of inducing variables exceeds $m_n \gtrsim n^{\frac{1}{3+2\beta}}$, i.e. for arbitrary $M_n \to \infty$*

$$E_{f_0}\Psi^* \left[ \|f - f_0\|_{L_2[0,1]} \geq M_n n^{-\beta/(3+2\beta)} \right] \to 0.$$

## 3.2 Heat equation

Next let us consider the problem of recovering the initial condition for the heat equation. The heat equation is often considered as the starting example in the PDE literature and, for instance, the Black-Scholes PDE can be converted to the heat equation as well. We consider the Dirichlet boundary condition

$$\frac{\partial}{\partial t}u(x,t) = \frac{\partial^2}{\partial x^2}u(x,t), \quad u(x,0) = \mu(x), \quad u(0,t) = u(1,t) = 0, \tag{12}$$

for $u$ defined on $[0,1] \times [0,T]$, $T > 0$. For $\mu \in L_2[0,1]$, $u(x,t) = \sqrt{2}\sum_{j=1}^{\infty}\mu_j e^{-j^2\pi^2 t}\sin(j\pi x)$, with $\mu_j = \sqrt{2}\int_0^1 \mu(s)\sin(j\pi s)ds$. Therefore, if $\mathcal{A} : \mathcal{D} \mapsto \mathcal{D}$, with $\mathcal{D} := \{f \in L_2[0,1], \ f(0) = f(1) = 0\}$, is such that, for $\mu = f$, $\mathcal{A}f(x) = u(x,T)$, then the corresponding singular-values and singular-functions of the operator $\mathcal{A}$ are $\kappa_j = e^{-j^2\pi^2 T}$ and $e_j(x) = g_j(x) = \sqrt{2}\sin(j\pi x)$. Therefore it is a severely ill-posed problem with $p = 2$ and $c = \pi^2 T$. We also note that $\sup_j \|e_j\|_\infty \vee \|g_j\|_\infty < \infty$. This problem has been well studied both in the frequentist [6; 19; 32] and Bayesian setting [28; 54]. Then, by direct application of Corollary 1 we can provide optimality guarantees for the variational Bayes procedure in this model as well.

**Corollary 3.** *Consider the heat equation operator $\mathcal{A}$ as above in the linear inverse regression model* (1) *and assume that $f_0 \in \bar{H}^\beta$ for some $\beta > 0$. Furthermore, we set the eigenvalues $\lambda_j = j^{-\alpha}e^{-\xi j^2}$, $\alpha \geq 0$, $\xi > 0$ in* (4). *Then the variational approximation $\Psi^*$ resulting from either of the spectral features inducing variables method with $m_n \geq \left(\xi + \pi^2 T\right)^{-1/2}\log^{1/2}n$ achieves the minimax contraction rate, i.e. for arbitrary $M_n \to \infty$*

$$E_{f_0}\Psi^*\left[\|f - f_0\|_{L_2([0,1])} \geq M_n \log^{-\beta/2}n\right] \to 0.$$

## 3.3 Radon transform

Finally, we consider the Radon transform [24], where for some (Lebesgue)–square-integrable function $f : D \to \mathbb{R}$ defined on the unit disc $D = \left\{x \in \mathbb{R}^2 : \|x\|_2 \leq 1\right\}$, we observe its integrals along any line intersecting $D$. If we parameterized the lines by the length $s \in [0,1]$ of their perpendicular from the origin and the angle $\phi \in [0.2\pi)$ of the perpendicular to the x-axis, we observe

$$\mathcal{A}f(s,\phi) = \frac{\pi}{2\sqrt{1-s^2}}\int_{-\sqrt{1-s^2}}^{\sqrt{1-s^2}} f(s\cos\phi - t\sin\phi, s\sin\phi + t\cos\phi)dt, \tag{13}$$

where $(s,\phi) \in S = [0,1] \times [0,2\pi)$. The Radon transform is then a map from $\mathcal{A}: L_2(D;\mu) \to L_2(S;G)$, where $\mu$ is $\pi^{-1}$ times the Lebesgue measure and $dG(s,\phi) = 2\pi^{-1}\sqrt{1-s^2}dsd\phi$. Then $\mathcal{A}$ is a bijective, mildly ill-posed linear operator of order $p = 1/4$. Furthermore, the operator's singular value decomposition can be computed via Zernike polynomials $Z_m^k$ (degree $m$, order $k$) and Chebyshev polynomials of the second kind $U_m(\cos\theta) = \sin\left((m+1)\theta\right)/\sin\theta \leq m+1$, see [24]. Translating it to the single index setting, we get for some functions $l,m : \mathbb{N} \mapsto \mathbb{N}$ satisfying $m(j) \asymp \sqrt{j}$ and $|l(j)| \leq m(j)$, that

$$e_j(r,\theta) = \sqrt{m(j)+1}Z_{m(j)}^{|l(j)|}e^{jl(j)\theta}, \quad g_j(s,\phi) = U_{m(j)}(s)e^{jl(j)\phi},$$

if polar coordinates are used on $D$. Therefore, we have $\sup_j(\|e_j\|_\infty \vee \|g_j\|_\infty)/\sqrt{j} < \infty$. Then, by directly applying Corollary 1 to this setting we can show that the variational Bayes method achieves the minimax contraction rate.

**Corollary 4.** *Consider the Radon transform operator* (13) *in the inverse regression model* (1) *and let us take $f_0 \in \bar{H}^\beta$, $\beta > 9/4$. Taking polynomially decaying eigenvalues $\lambda_j \asymp j^{-1-2\beta}$, the empirical spectral features variational Bayes method achieves the optimal minimax contraction rate if $m_n \gtrsim n^{1/(3/2+2\beta)}$, i.e. for any $M_n \to \infty$*

$$E_{f_0}\Psi^*\left[\|f - f_0\|_{L_2(D;\mu)} \geq M_n n^{-\beta/(3/2+2\beta)}\right] \to 0.$$

# 4 Numerical analysis

We demonstrate the approximation accuracy of the variational Bayes method on synthetic data. We consider here the recovery of the initial condition of the heat condition 3.2, which is a severely

ill-posed. In the supplement we provide additional simulation study for mildly ill-posed inverse problems as well. We set the sample size $n = 8000$, take uniformly distributed covariates on $[0, 1)$, and let

$$f_0(t) = \sqrt{2} \sum_j c_j j^{-(1+\beta)} \sin(j\pi t), \quad c_j = \begin{cases} 1 + 0.4\sin(\sqrt{5}\pi j), & j \text{ odd}, \\ 2.5 + 2\sin(\sqrt{2}\pi j), & j \text{ even}, \end{cases}$$

for $\beta = 1$. The independent- observations are generated as $Y_i \sim \mathcal{N}(\mathcal{A}f_0(x_i), 1)$, depending on the solution of the forward map $\mathcal{A}f_0$ after time $T = 10^{-2}$.

We consider the prior with $\lambda_j = e^{-\xi j^2}$ for $\xi = 10^{-1}$. In view of Corollary 3 the optimal number of inducing variables is $m = (\xi + 2\pi^2 T)^{-1/2} \log^{1/2} n \approx 6$. We consider the population spectral feature method described in (10) and plot the variational approximation of the posterior for $m = 6$ and $m = 3$ inducing variables in Figure 1. We represent the true posterior mean by solid red and the upper and lower pointwise $2.5\%$ quantiles by dashed red curves. The true function is given by blue and the mean and quantiles of the variational approximation by solid and dotted purple curves, respectively.

Observe that with $m = 6$, see left part of Figure 1, the variational approximation results in similar $95\%$ pointwise credible bands and posterior mean as the true posterior, providing an accurate approximation. Also note that both the true and the variational posterior contain $f_0$ at most of the points, indicating frequentist confidence validity of the set. At the same time, by taking a factor of two less inducing points, i.e. $m = 3$, the credible sets will be overly large, resulting in good frequentist coverage, but suboptimally large posterior spread, see the second plot in Figure 1.

The computations were carried out with a 2,6 GHz Quad-Core Intel Core i7 processor. The computation of the exact posterior mean and covariance kernel on a grid of $100$ points took over half an hour (in CPU time), while the variational approximation was substantially faster, taking only $50.5$ ms, resulting in a $3.68 * 10^4$ times speed.

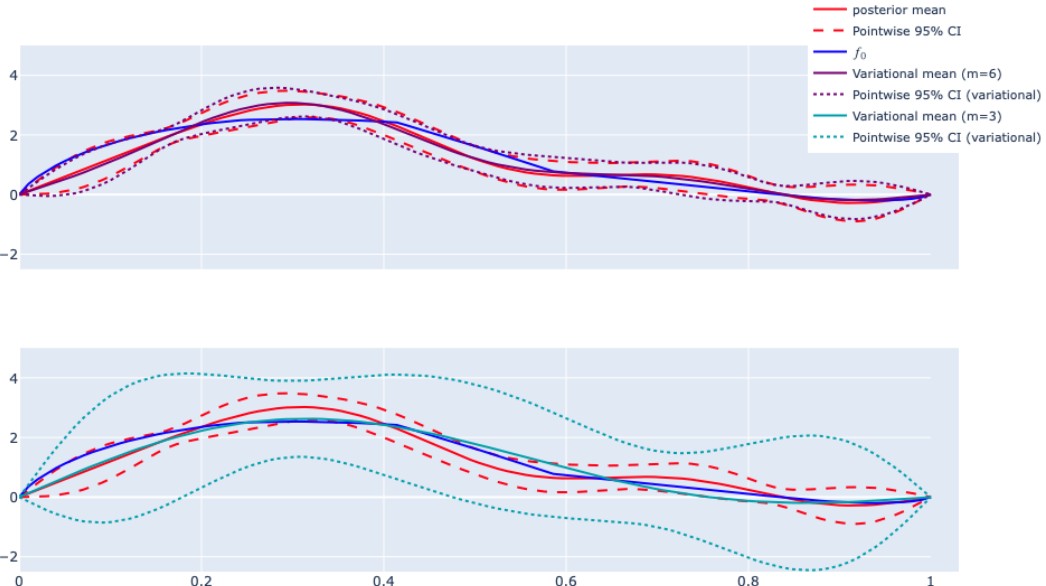

Figure 1: True and variational posterior means and credible regions for Gaussian series prior (sine basis) on the initial condition $\mu = f_0$ of the heat equation (12), for $m = 6$ (left) or $m = 3$ (right) inducing variables from method (10).

A more extensive numerical analysis is available in the appendix, considering the application of our method to the settings of Sections 3.1 and 3.3 as well. We conduct these experiments several times and compare the average Mean Integrated Squared Error (MISE), see appendix A, and compute time for different choices of $m$. We observe that in all our examples, while increasing $m$ results in longer computation, the MISE does not improve after a threshold close to the one presented in our results.

Therefore, it is sufficient to include as many inducing variables as we considered in Corollary 1 to obtain better performance. More than that would would only increase the computation complexity. In the Appendix, we also provide a literature review and some justifications of how relevant these problems are in practice

## 5   Discussion

We have extended the inducing variables variational Bayes method for linear inverse problems and derived asymptotic contraction rate guarantees for the corresponding variational posterior. Our theoretical results provide a guide for practitioners on how to tune the prior distribution and how many inducing variables to apply (in the spectral feature variational Bayes method) to obtain minimax rate optimal recovery of the true functional parameter of interest. We have demonstrated the practical relevance of this guideline numerically on synthetic data and have shown that using less variables results in highly suboptimal recovery.

In our analysis we have considered priors built on the singular basis of the operator $\mathcal{A}$. In principle our results can be extended to other priors as well, until the eigenbasis of the covariance operator is not too different from the basis of the operator $\mathcal{A}$. This, however, would complicate the computation of the Kullback -Leibler divergence between the variational family and the posterior, resulting in extra technical challenges. In this setting the empirical spectral features method seems practically more feasible, especially, if the eigenbasis of the covariance kernel is not known explicitly. In the literature several different types of inducing variable methods were proposed, considering other, practically more relevant approaches of interest. Furthermore, extension to other type of inverse problems is also feasible. For instance in the deconvolution problem, when $f_0$ is convoluted with a rectangular kernel, the eigenvalues given by the SVD are the product of a polynomially decaying and oscillating part and the "average degree" of ill-posedness does not match the lower and upper bounds [23]. Extension to non-linear inverse problem is highly relevant, as these problems tend to be computationally even more complex, but very challenging. One possible approach is to linearize the problem and take its variational approximation. Finally, it is of importance to derive frequentist coverage guarantees for VB credible sets. Our approach cannot directly be extended for this task. However, in the direct case, for some special choices of inducing variables, frequentist coverage guarantees were derived using kernel ridge regression techniques [38; 60]. This result, although computationally somewhat cumbersome, in principle can be extended to the inverse setting as well. One last drawback of our results is that the priors we consider are non-adaptative in the mildly ill-posed case. Minimax contraction rates are attainable only if the covariance eigenvalues are properly tuned, given the smoothness $\beta$. While this is not an issue the severely ill-posed case in our results, we keep the study of adaptation for future works as it is a much more involved question.

*Funding.* Co-funded by the European Union (ERC, BigBayesUQ, project number: 101041064). Views and opinions expressed are however those of the author(s) only and do not necessarily reflect those of the European Union or the European Research Council. Neither the European Union nor the granting authority can be held responsible for them.

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
