# Variational Gaussian Processes For Linear Inverse Problems:
# Supplementary material

**Thibault Randrianarisoa**
Department of Decision Sciences
Bocconi University
via Roentgen 1, 20136, Milano, MI, Italy
`thibault.randrianarisoa@unibocconi.it`

**Botond Szabo**
Department of Decision Sciences
Bocconi University
via Roentgen 1, 20136, Milano, MI, Italy
`botond.szabo@unibocconi.it`

## A  Additional experiments

### A.1  Heat equation

Pursuing the study of the recovery of the initial condition of the heat equation of Section 4, we repeat the experience 50 times with $n = 4000$ observations, considering all other parameters identical to those used before. According to our theory, the number of inducing variables we should use is still equal to $m = \left(\xi + 2\pi^2 T\right)^{-1/2} \log^{1/2} n \approx 6$. As before, we consider the population spectral feature method described in (10).

The results from one experiment are presented in Figure 2. We plot the resulting variational approximation of the posterior for $m = 6$ and $m = 3$ inducing variables and represent the true posterior mean by solid red and the upper and lower pointwise $2.5\%$ quantiles by dashed red curves. The true function is given by blue and the mean and quantiles of the variational approximation by solid and dotted purple/cyan curves, respectively.

The conclusions we draw from this experiment are the same as those in Section 4. With the optimal choice $m = 6$ following from our theoretical results(on the left of Figure 2), the posterior and variational means are almost indistinguishable and the 95% pointwise credible bands are identical and contains $f_0$ almost everywhere. However, reducing the number of inducing points to $m = 3$, the variational credible sets become much larger, providing less information about $f_0$, and the variational posterior mean is smoother, providing a worse fit to $f_0$.

In Figure 3, we summarize the results from the 50 experiments we ran, assessing the quality of the different posterior distributions we consider via the mean integrated squared error (MISE)

$$\int \|f - f_0\|^2_{L_2(\mathcal{T};\mu)} \, d\Pi[f|X, Y],  \tag{14}$$

which can be computed explicitly as the posterior is Gaussian. We compare the true posterior and the variational posteriors obtained with the optimal choice of $m = 6$ inducing variables and twice more/less variables with $m = 12$ and $m = 3$, respectively. On the right-hand side of Figure 3, we see that $m = 3$ is a suboptimal choice as it results in a much higher MISE than the other approaches. On the left-hand side of Figure 3, we also report the computation times of the methods, and we highlight that the true posterior takes much longer than any of the variational approximations. On Figure 4, we further see that increasing the number of inducing variables results in more computation time, as expected. At the same time, increasing the number of inducing variables beyond the optimal threshold ($m = 6$) does not increase the accuracy considerably.

37th Conference on Neural Information Processing Systems (NeurIPS 2023).

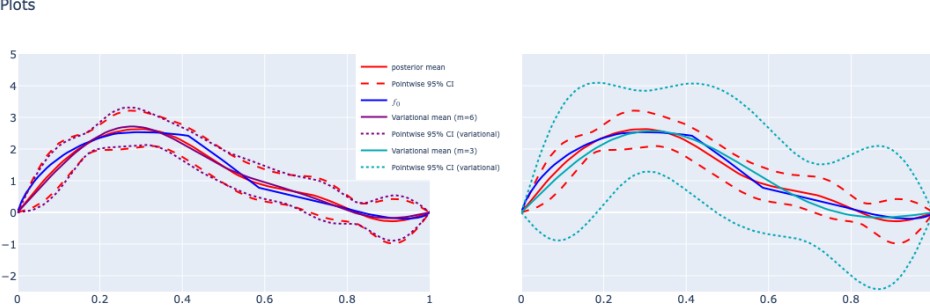

Figure 2: True and variational posterior means and credible regions for Gaussian series prior (sine basis) on the initial condition $\mu = f_0$ of the heat equation (12), with $m = 6$ (left) or $m = 3$ (right) inducing variables from method (10), computed from $n = 4000$ observations.

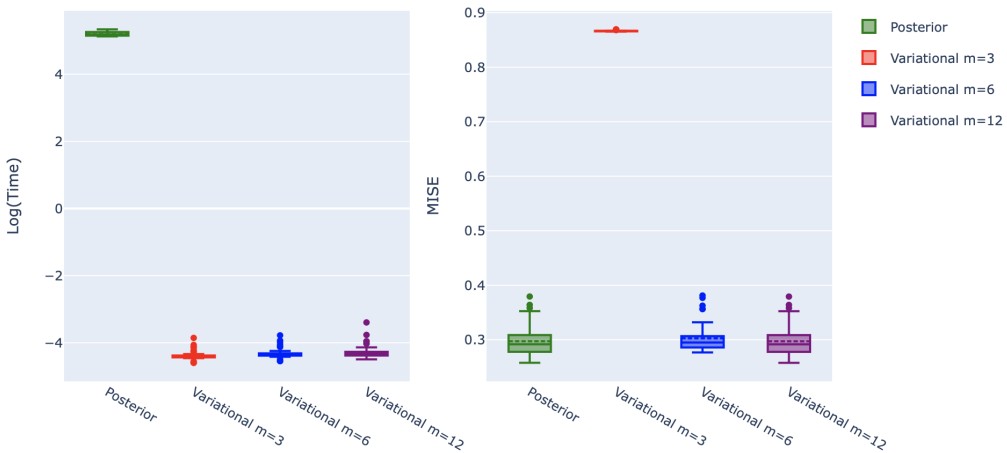

Figure 3: Boxplots of the (logarithm of) computation time (in seconds, on the left) and the MISE (on the right) of the true and variational posteriors for Gaussian series prior (sine basis) on the initial condition $\mu = f_0$ of the heat equation (12), with $m = 3, 6, 12$ inducing variables from method (10), obtained from 50 experiments with $n = 4000$ samples.

### A.2 Volterra operator

Next we consider the Volterra operator (11). This is a mildly ill-posed problem of degree $p = 1$. We set the sample size $n = 15000$, take uniformly distributed covariates on $[0, 1]$, and let

$$f_0(t) = \sqrt{2} \sum_j c_j j^{-(1+\beta)} \cos((j - 1/2)\pi t), \quad c_j = \begin{cases} 1 + 0.9 \sin(\sqrt{3}\pi j), & j \text{ odd}, \\ 1 + 0.8 \sin(\sqrt{7}\pi j), & j \text{ even}, \end{cases}$$

for $\beta = 0.6$, so that $f_0 \in \bar{H}^\beta$. The independent observations are then generated as $Y_i \sim \mathcal{N}(\mathcal{A}f_0(x_i), 1)$, depending on the primitive of $f_0$.

We consider the prior with $\lambda_j = j^{-1-2\beta}$. In view of Corollary 3 the optimal number of inducing variables is $m = n^{\frac{1}{3+2\beta}} \approx 10$. We consider the population spectral features method described in (10) and plot the variational approximation of the posterior for $m = 10$ and $m = 5$ inducing variables in Figure 5, using the same colorcode as in the previous section.

With $m = 10$ on the top of Figure 5, the variational approximation results in similar 95% pointwise credible bands and posterior mean as the true posterior, providing an accurate approximation. On the bottom of this figure, we observe that the mean and pointwise credible bands with $m = 5$ inducing variables are considerably different, though in both cases, the credible bands contain $f_0$.

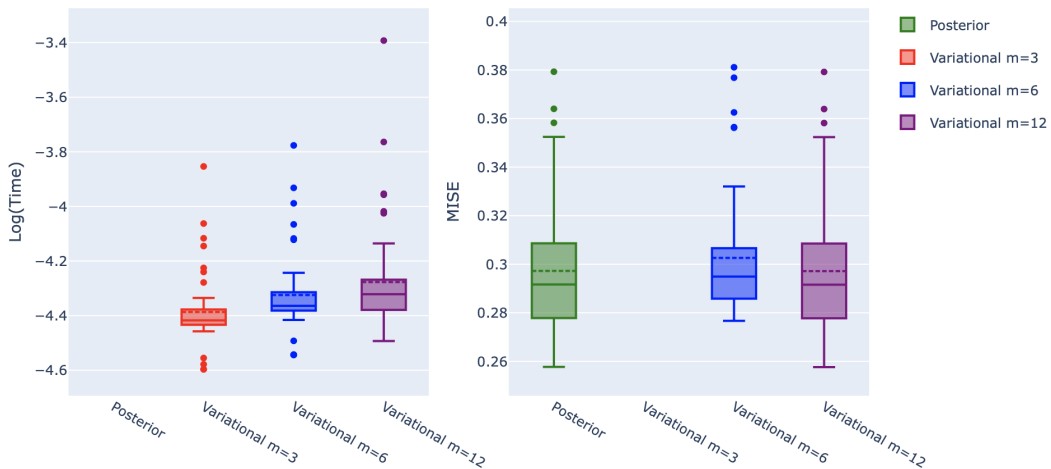

Figure 4: Zoom of Figure 3.

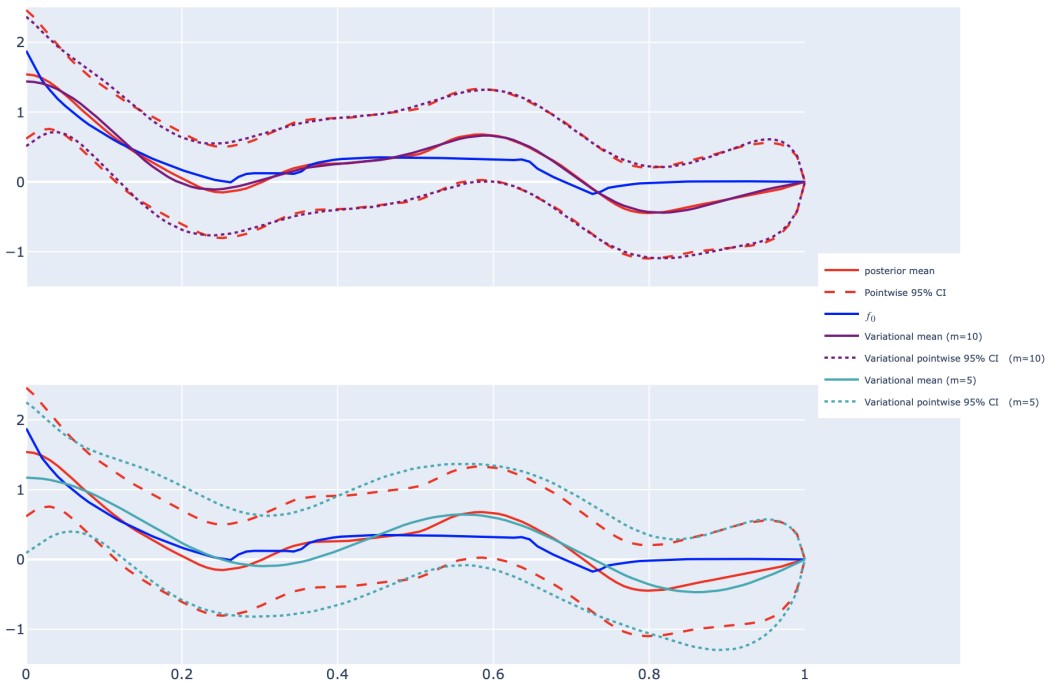

Figure 5: True and variational posterior means and credible regions for Gaussian series prior (cosine basis) and $m = 10$ (top) or $m = 5$ (bottom) inducing variables from method (10) for the Volterra operator in Section 3.1.

We repeat the above experiment 30 times with $n = 4000$ and compare the computation times and MISE (14) of the true posterior and the variational posteriors obtained with the optimal choice of $m = 8$ inducing variables and twice more/less variables $m = 16/m = 4$. Looking at Figures 6 and 7, the same message holds as before, in case of the heat equation.

We also illustrate and compare theoretical and empirical phase-transition curves on synthetic data coming from the Volterra operator (11). We computed the (logarithm of the) ratio of the mean integrated squared error (MISE) corresponding to the true and variational posteriors (we simulate 20 experiments each time to empirically approximate these quantities). We have considered $n$ ranging from 100 to 10000 and $m$ from 1 to 17, under the same setting as above. We have also plotted

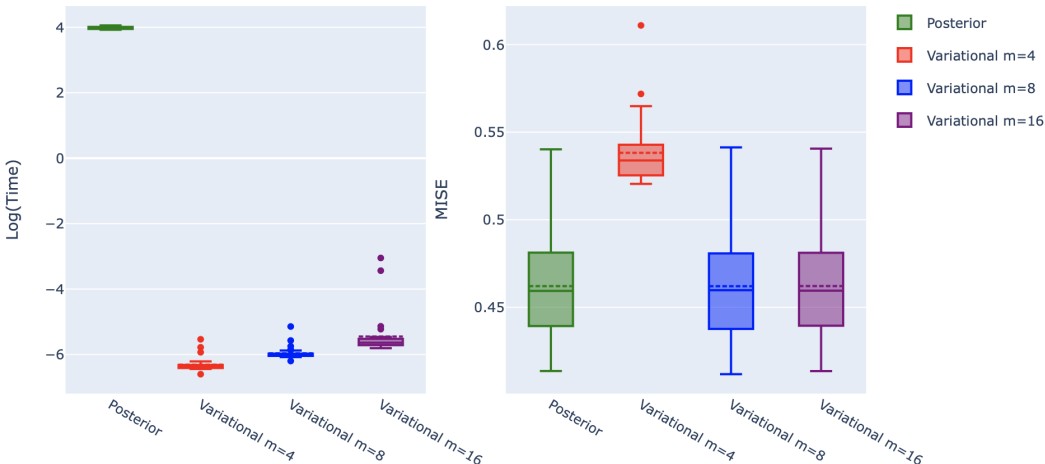

Figure 6: Boxplots of the (logarithm of) computation time (in seconds, on the left) and the MISE (on the right) of the true and variational posteriors for Gaussian series prior (cosine basis) with $m = 4, 8, 16$ inducing variables from method (10), obtained from 50 experiments with $n = 4000$ samples, for the Volterra operator in Section 3.1.

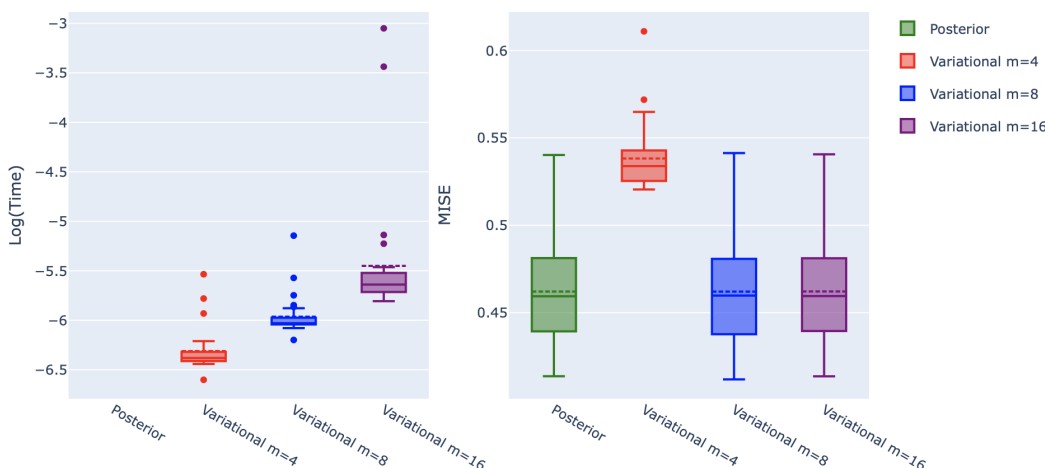

Figure 7: Zoom of Figure 6.

the phase transition curve (white line) coming from our theoretical analysis on Figure 8. One can note that the theoretical curve closely resembles the curve where the phase transition occurs in the empirical study. Indeed, there is not much empirical improvement of the MISE after the threshold given by Corollary 2.

## A.3 Radon transform

### A.3.1 Experiments

We now turn to a simulation study of the Radon transform (13), which represents a mildly ill-posed problem of degree $p = 1/4$. We observe the performance of the true posterior and variational approximations for different sample sizes, $n = 500$ and $n = 5000$, and take independent covariates drawn from the distribution $dG(s, \phi) = 2\pi^{-1}\sqrt{1 - s^2}dsd\phi$ on $S = [0, 1] \times [0, 2\pi]$. We set the polar coordinates of the functional parameter $f_0$ on the unit disc $D = \{x \in \mathbb{R}^2 : \|x\|_2 \leq 1\}$ as

$$f_0(r, \theta) = \sum_j c_j j^{-(1+\beta)} e_j(r, \theta), \quad c_j = \begin{cases} 1 + 0.5 \sin(\sqrt{3}\pi j), & j \text{ odd}, \\ 2 + 0.8 \sin(\sqrt{7}\pi j), & j \text{ even}, \end{cases}$$

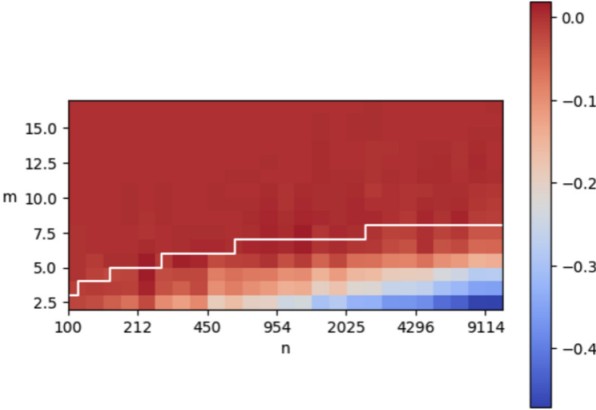

Figure 8: Log ratio $\log \frac{E_{f_0} \mathrm{MISE}(\Pi[\cdot|X,Y])}{E_{f_0} \mathrm{MISE}(\Psi^*)}$ of MISE between the true and variational posteriors recovering $f_0$ from its image by the Volterra operator in Section 3.1. In white is represented the function $n \to \lceil n^{\frac{1}{3+2\beta}} \rceil$ given by Corollary 2.

for $\beta = 0.6$ and

$$
e_j(r,\theta) = \begin{cases} Z_{m_j}^{|l_j|}(r)\cos(l_j\theta), & l_j > 0, \\ Z_{m_j}^{|l_j|}(r), & l_j = 0, \\ Z_{m_j}^{|l_j|}(r)\sin(l_j\theta), & l_j < 0, \end{cases}
$$

$m_j = \lceil \frac{\sqrt{1+8j}-1}{2} \rceil - 1$ and $l_j = 2(j-1) - m_j(m_j+2)$. Note that $f_0 \in \bar{H}^\beta$. The independent observations are again generated as $Y_i \sim \mathcal{N}(\mathcal{A}f_0(x_i), 1)$.

We again consider the prior eigenvalues $\lambda_j = j^{-1-2\beta}$. The optimal number of inducing variables is $m = n^{\frac{2}{3+4\beta}}$, which for the different sample sizes $n = 500$ and $n = 5000$ we consider is equal to 10 and 24, respectively. We consider the population spectral feature method described in (10) and plot the variational approximation of the posterior for $m$ and $\lceil m/4 \rceil$ inducing variables in Figure 9 ($n = 500$) and Figure 10 ($n = 5000$). For each setting, we represent the true posterior mean, the variational means, the upper and lower pointwise $2.5\%$ quantiles as well as the absolute pointwise difference between $f_0$ and the posterior/variational means $\hat{f}_n$. Negative values are represented in blue, positive ones in red and points corresponding to small absolute values are in white.

Again, similar conclusions can be drawn as in the previous sections. Observe that the true posterior and variational means are similar for the optimal choice of $m$, while choosing four times less inducing variables results in a posterior mean that is much smoother. On the other hand, the credible bands with the suboptimal choice of $m$ are overly large compared to the true posteriors.

### A.3.2 Applications

The Radon transform is a mathematical technique with various applications, particularly in the field of medical imaging and image analysis. It is used to analyze and transform data from the spatial domain to the Radon domain, providing a different perspective on the data that can be useful for specific tasks. Inverting the Radon transform has found a lot of applications where lower-dimensional integrals of the inside of an object are more readily available than the object of interest itself. We provide below a non-exhaustive list of possible applications:

- *Computed Tomography (CT) Imaging and Medical Single Photon Emission Computed Tomography (SPECT)*: In CT scans, X-ray measurements are taken from different angles around a patient, and the Radon transform is used to reconstruct a cross-sectional image (slice) of the patient's body. This helps doctors visualize internal structures and diagnose various medical conditions. SPECT is a nuclear medicine imaging technique that uses gamma-ray detectors to generate 3D images of the distribution of radioactive tracers within a

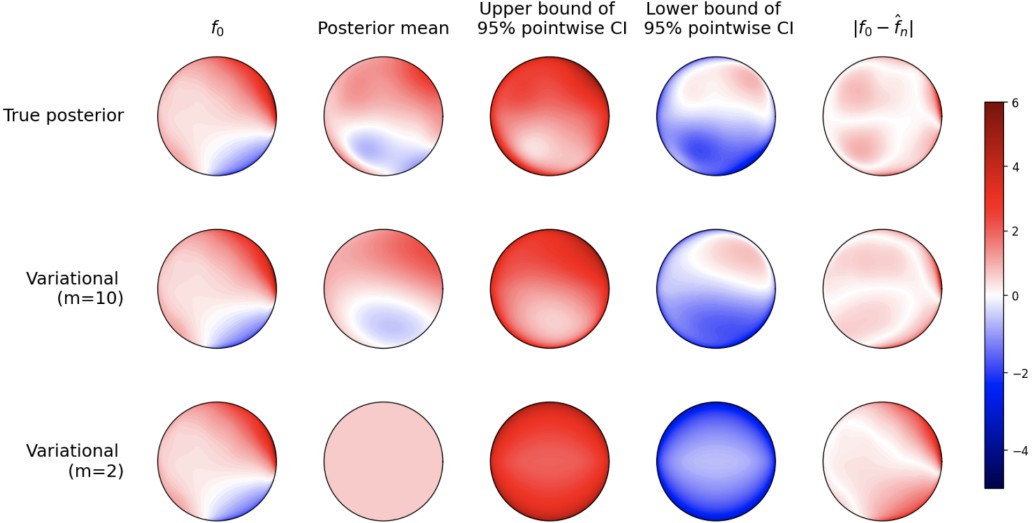

Figure 9: True and variational posterior means and credible regions for Gaussian series prior (Zernike polynomial basis) and $m = 17$ (middle) or $m = 8$ (bottom) inducing variables from method (10), with $n = 5000$, recovering the parameter $f_0$ from its Radon transform (see Section 3.3).

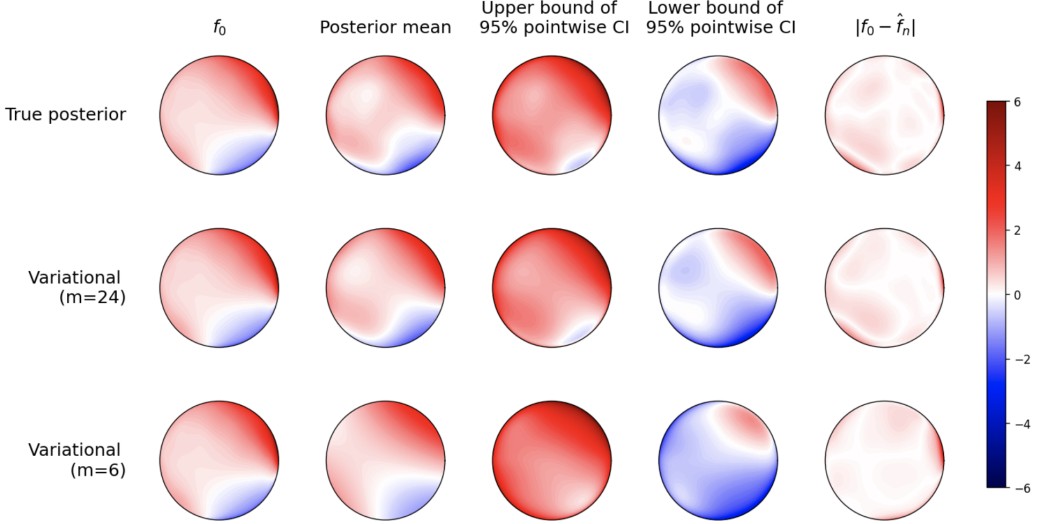

Figure 10: True and variational posterior means and credible regions for Gaussian series prior (Zernike polynomial basis) and $m = 24$ (middle) or $m = 12$ (bottom) inducing variables from method (10), with $n = 5000$, recovering the parameter $f_0$ from its Radon transform (see Section 3.3).

patient's body. The Radon transform is used in the image reconstruction process for SPECT. [3; 5; 45; 55]

- *Seismic Imaging*: In seismology, the Radon transform is employed to process seismic data collected from earthquakes or controlled explosions. It helps create images of the subsurface structure of the Earth, aiding in oil and gas exploration and understanding geological formations. [18; 19]

- *Geophysical Imaging*: The Radon transform has applications in geophysical imaging techniques such as ground-penetrating radar (GPR), where it helps in image reconstruction to understand subsurface properties. [43]

- *Radar and Sonar Imaging*: The Radon transform is used in underwater sonar imaging to reconstruct images of underwater objects or terrains, or radar imaging to create imaging of landscapes. This has applications in marine biology, naval operations, and underwater exploration.[54; 59]

- *Particle Tracking*: In high-energy physics and particle physics experiments, the Radon transform is used to analyze data from particle detectors to track the paths of particles, determining their trajectories and energies.[40]

- *Material Science and Crystallography*: The Radon transform can be applied to analyze diffraction patterns in crystallography and material science, helping to understand the structure of materials at the atomic level.[6]

In Mathematics, the Radon transform has also been used to solve hyperbolic partial differential equations via the method of plane waves, which reduces the problem to the resolution of ordinary differential equations [67; 60].

# B  Proof of Theorem 1

We start by introducing some notation and background information used throughout the proof. First note, that since the eigenfunctions of the covariance kernel $k$ were chosen to be the eigenfunctions of $\mathcal{A}^*\mathcal{A}$, the prior $\Pi_\mathcal{A}$ on $\mathcal{A}f$, induced by the GP prior $\Pi$ on $f$, is also a centered Gaussian process with covariance kernel

$$(x,y) \mapsto \sum_{j=1}^\infty \lambda_j \kappa_j^2 g_j(x) g_j(y), \tag{15}$$

i.e. the eigenvalues and eigenfunctions of the kernel are $(\lambda_j \kappa_j^2)_{j \in \mathbb{N}}$ and $(g_j)_{j \in \mathbb{N}}$, respectively. Let us denote by $\mathbb{H}_\mathcal{A}$ the corresponding Reproducing Kernel Hilbert Space (RKHS) and by $\mathbb{H}$ the RKHS corresponding to the prior $\Pi$ on $f$. In view of Theorem I.18 of [23], the above RKHS takes the form

$$\mathbb{H}_\mathcal{A} = \left\{ h(x) = \sum_{j=1}^\infty h_j g_j(x) \colon \ \|h\|_{\mathbb{H}_\mathcal{A}}^2 := \sum_{j=1}^\infty h_j^2 \lambda_j^{-1} \kappa_j^{-2} < \infty \right\}, \tag{16}$$

where $h_j = \langle h, g_j \rangle_{L_2(\mathcal{X};G)}$. Furthermore, note that for all measurable set $S \subset L_2(\mathcal{X};G)$ we have for $Z_j \sim^{iid} N(0,1)$ random variables, that

$$\Pi(f\colon \ \mathcal{A}f \in S) = P(\sum_{j=1}^\infty \lambda_j^{1/2} \kappa_j Z_j g_j \in S) = \Pi_\mathcal{A}(w\colon \ w \in S). \tag{17}$$

In the next sections, we denote the rates for the direct problems by

$$\varepsilon_n = M \begin{cases} n^{-\frac{\alpha \wedge \beta + p}{1+2\alpha+2p}} & \text{in the mildly ill-posed case,} \\ n^{-c/(\xi+2c)} \log^{-\beta/p+c\alpha/(\xi+2c)}(n) & \text{in the severely ill-posed case,} \end{cases} \tag{18}$$

for some $M > 0$ large enough.

Finally, we provide an explicit formula of the KL divergence between the posterior distribution $\Pi[\cdot \,|X,Y]$ and the variational approximation $\Psi^*$. It can be expressed with the evidence lower bound $\mathcal{L}$ as

$$\mathrm{KL}\left(\Psi^*\|\Pi[\cdot|X,Y]\right) = \log p(X,Y) - \mathcal{L},$$

where computations from [74] give

$$\mathcal{L} := \log \int \exp \Big( \int \log p_f(X,Y) d\Pi(f|\mathbf{u}) \Big) d\Pi_u(\mathbf{u})$$

$$= -\big|2\pi(\sigma^2 I_n + K_{\mathcal{A}f\mathcal{A}f})\big| - \frac{1}{2\sigma^2}\mathbf{y}\left(\sigma^2 I_n + K_{\mathcal{A}f\mathcal{A}f}\right)^{-1}\mathbf{y} - Tr\left(K_{\mathcal{A}f\mathcal{A}f} - Q_{\mathcal{A}f\mathcal{A}f}\right),$$

for $Q_{\mathcal{A}f\mathcal{A}f} = K_{\mathcal{A}fu} K_{uu}^{-1} K_{u\mathcal{A}f}$. Then the KL divergence takes the form

$$\mathrm{KL}\left(\Psi^*\|\Pi[\cdot|X,Y]\right) = \frac{1}{2}\Big( \mathbf{y}\big[ \left(\sigma^2 I_n + Q_{\mathcal{A}f\mathcal{A}f}\right)^{-1} - \left(\sigma^2 I_n + K_{\mathcal{A}f\mathcal{A}f}\right)^{-1} \big] \mathbf{y}^T$$

$$+ \log \frac{|\sigma^2 I_n + Q_{\mathcal{A}f\mathcal{A}f}|}{|\sigma^2 I_n + K_{\mathcal{A}f\mathcal{A}f}|} + \frac{1}{\sigma^2} Tr(K_{\mathcal{A}f\mathcal{A}f} - Q_{\mathcal{A}f\mathcal{A}f}) \Big). \tag{19}$$

## B.1  Step 1: Empirical $L_2$ contraction in the direct problem

As a first step we fix the design points and derive posterior contraction rate around $\mathcal{A}f_0$ with respect to the empirical $L_2(\mathcal{X};P_n)$-norm, i.e. $\|w\|_{L_2(\mathcal{X};P_n)}^2 = n^{-1}\sum_{i=1}^n w(x_i)^2$. More precisely, we show that there exists an event $A_n$ with $P_X(A_n) \to 1$ and events $B_{n,|X}$ conditional on the design $X$ with $\inf_{X \in A_n} P_{Y\,|\,X}(B_{n,|X}) \to 1$, such that for any sequence $M_n \to \infty$ and $X \in A_n$

$$E_{Y\,|\,X} \Pi \left[ f\colon \|\mathcal{A}f - \mathcal{A}f_0\|_{L_2(\mathcal{X};P_n)} \geq M_n \varepsilon_n \mid X, Y \right] \mathbb{1}_{B_{n,|X}} \leq C e^{-cM_n^2 n \varepsilon_n^2} \tag{20}$$

holds for $\varepsilon_n$ given in (18).

Let us recall the definition of the concentration function (in case of the direct problem)

$$\phi_{\mathcal{A}f_0}(\varepsilon) := \inf_{h \in \mathbb{H}_{\mathcal{A}}: \|\mathcal{A}f_0 - h\|_{L_2(\mathcal{X};P_n)} < \varepsilon} \|h\|_{\mathbb{H}_{\mathcal{A}}}^2 - \log \Pi_{\mathcal{A}}\left(w: \|w\|_{L_2(\mathcal{X};P_n)} < \varepsilon\right).$$

Then in view of Theorem 3.3 of [78], to prove (20) it is sufficient to verify the concentration inequality

$$\phi_{\mathcal{A}f_0}(\varepsilon_n) \leq n\varepsilon_n^2. \tag{21}$$

This result is based on [22] where in the proof it is shown that there exists a sequence of events $B_{n,|X}$ such that $\sup_X P_{Y|X}\left(B_{n,|X}^c\right)$ vanishes and (20) holds $P_X$-almost surely.

We prove (21) in two steps. First we verify it for the $L_2(\mathcal{X};G)$-norm, i.e. we show that for $M$ large enough in (18),

$$\inf_{h \in \mathbb{H}_{\mathcal{A}}: \|h - \mathcal{A}f_0\|_{L_2(\mathcal{X};G)} \leq \varepsilon_n} \|h\|_{\mathbb{H}_{\mathcal{A}}}^2 \leq n\varepsilon_n^2, \tag{22}$$

$$- \log \Pi_{\mathcal{A}}\left(w: \|w\|_{L_2(\mathcal{X};G)} < \varepsilon_n\right) \leq n\varepsilon_n^2. \tag{23}$$

Then we relate the population $L_2(\mathcal{X};G)$-norm to the empirical $L_2(\mathcal{X};P_n)$-norm on a large enough event $A_n$, finishing up the argument. We note that one can not apply this result to the $L_2(\mathcal{X};G)$-norm as the testing metric (Hellinger) and the $L_2$-norm do not satisfy the required connection.

In the mildly ill-posed case the above inequalities directly follow from Lemma 1 and 2, respectively. In the severely ill-posed case for (22) in view of Lemma 1 it is sufficient to verify that $J_{\varepsilon_n}^{\alpha-2\beta} e^{\xi J_{\varepsilon_n}^p} \leq n\varepsilon_n^2$ for $J_{\varepsilon_n}^{\beta} e^{cJ_{\varepsilon_n}^p} \asymp \varepsilon_n^{-1}$. Note that by substituting $\varepsilon_n$ in the previous inequality, we equivalently get $J_{\varepsilon_n}^{\alpha} e^{(\xi+2c)J_{\varepsilon_n}^p} \lesssim n$. Then, in view of Section 3.3 of [32] (using the Lambert function) this holds for some $J_{\varepsilon_n} = O(\log^{1/p} n)$. Furthermore, following from $e^{J_{\varepsilon_n}^p} \lesssim \left(nJ_{\varepsilon_n}^{-\alpha}\right)^{1/(\xi+2c)}$, we arrive at

$$\varepsilon_n \asymp J_{\varepsilon_n}^{-\beta} e^{-cJ_{\varepsilon_n}^p} \gtrsim n^{-c/(\xi+2c)} \log^{-\beta/p+c\alpha/(\xi+2c)}(n),$$

finishing the proof of (22). For (23), in view of Lemma 2, we need $\varepsilon_n \gtrsim n^{-1/2} \log^{(p+1)/2p} n$, which holds for $\varepsilon_n$.

It remained to replace in (22) and (23) the $L_2(\mathcal{X};G)$-norm with the $L_2(\mathcal{X};P_n)$-norm. First note that in view of Lemma 5, there exists an event $A_{n,1}$ with $P_X(A_{n,1}^c) = o(1)$ such that for $X \in A_{n,1}$

$$\Pi_{\mathcal{A}}\left(\|w\|_{L_2(\mathcal{X};P_n)} < C\varepsilon_n\right) \geq \Pi_{\mathcal{A}}\left(\|w\|_{L_2(\mathcal{X};G)} < \varepsilon_n\right) + o\left(e^{-n\varepsilon_n^2}\right) \gtrsim e^{-n\varepsilon_n^2}.$$

Furthermore, note that the upper bound in Lemma 1 were derived for $h = \mathcal{A}f_0^{J_\varepsilon}$ with appropriately chosen $J_\varepsilon$. Then in view of Lemma 8 (with $J = J_{\varepsilon_n} < k$ in the lemma) there exists an event $A_{n,2}$ with $P_X(A_{n,2}^c) = o(1)$ such that

$$\|h - \mathcal{A}f_0\|_{L_2(\mathcal{X};P_n)} = \|\mathcal{A}f_0^{\perp J_{\varepsilon_n}}\|_{L_2(\mathcal{X};P_n)} \lesssim \|\mathcal{A}f_0^{\perp J_{\varepsilon_n}}\|_{L_2(\mathcal{X};G)} + o(\varepsilon_n) \lesssim \varepsilon_n,$$

where $\mathcal{A}f_0^{\perp J}(x) = \sum_{j=J+1}^{\infty} \kappa_j f_{0,j} g_j(x)$ and we used that $(\alpha \wedge \beta) + p > 3/2 + 2\gamma$ in the first bound, verifying the statement on the event $A_n = A_{n,1} \cap A_{n,2}$ satisfying $P_X(A_n^c) = o(1)$, for some large $M > 0$.

## B.2 Step 2: Population $L_2$ contraction rate in the direct problem

Next we adapt the contraction rate result (20) to the random design regression model and consider $L_2(\mathcal{X};G)$ contraction rate, i.e. we show that there exists a sequence of events $D_n$ with $P_{X,Y}(D_n^c) = o(1)$ such that

$$E_{f_0}\Pi\left[f: \|\mathcal{A}f - \mathcal{A}f_0\|_{L_2(\mathcal{X};G)} \geq M_n \varepsilon_n \mid X, Y\right] \mathbb{1}_{D_n} \leq Ce^{-cM_n^2 n\varepsilon_n^2}. \tag{24}$$

First note that in view of Lemma 5 for $f \in \mathcal{F}_n$ defined in (33) we have on an event $A_{n,1}$ with $P_X(A_{n,1}^c) = o(1)$ that $\|\mathcal{A}f - \mathcal{A}f_0\|_{L_2(\mathcal{X};G)} \leq C(\|\mathcal{A}f - \mathcal{A}f_0\|_{L_2(\mathcal{X};P_n)} + \varepsilon_n)$. Furthermore, note that (22) and (23) in view of Proposition 11.19 of [23] imply that for some $c > 0$

$$\Pi[f: \|\mathcal{A}f - \mathcal{A}f_0\|_{L_2(\mathcal{X};G)} \leq \epsilon_n] \gtrsim e^{-cn\varepsilon_n^2}. \tag{25}$$

In view of $\Pi(f \in \mathcal{F}_n^c) \leq e^{-n^{\frac{1}{1+2\gamma}} n\varepsilon_n^2}$, see Lemma 5, Lemma 4 gives $\Pi(f \in \mathcal{F}_n^c | X, Y) \leq e^{-n^{\frac{1}{1+2\gamma}} n\varepsilon_n^2/2}$. Furthermore, in view of (20) there exists an event $A_{n,2}$ with $P_{X,Y}(A_{n,2}^c) = o(1)$ such that

$$E_X E_{Y|X} \Pi \left[ f \colon \|\mathcal{A}f - \mathcal{A}f_0\|_{L_2(\mathcal{X};P_n)} \geq M_n \varepsilon_n \mid X, Y \right] \mathbb{1}_{A_{n,2}} \lesssim e^{-cM_n^2 n\varepsilon_n^2}.$$

Therefore, by taking $D_n = A_{n,1} \cap A_{n,2}$ we get that

$$
\begin{aligned}
E_{f_0} \Pi & \left[ f \colon \|\mathcal{A}f - \mathcal{A}f_0\|_{L_2(\mathcal{X};G)} \geq M_n \varepsilon_n \mid X, Y \right] \mathbb{1}_{A_n} \\
& \leq E_{f_0} \Pi \left[ f \in \mathcal{F}_n \colon \|\mathcal{A}f - \mathcal{A}f_0\|_{L_2(\mathcal{X};G)} \geq M_n \varepsilon_n \mid X, Y \right] \mathbb{1}_{A_n} + E_{f_0} \Pi \left[ f \in \mathcal{F}_n^c \mid X, Y \right] \\
& \leq E_X \left( E_{Y|X} \Pi \left[ f \colon \|\mathcal{A}f - \mathcal{A}f_0\|_{L_2(\mathcal{X};P_n)} \geq CM_n \varepsilon_n \mid X, Y \right] \mathbb{1}_{A_n} \right) + e^{-n^{\frac{1}{1+2\gamma}} n\varepsilon_n^2/2} \\
& \lesssim e^{-\left( n^{\frac{1}{1+2\gamma}} \wedge M_n^2 \right) n\varepsilon_n^2/2}.
\end{aligned}
$$

## B.3  Step 3: Population $L_2$ contraction rate in the indirect problem

Next, we turn the contraction rate results for $\mathcal{A}f$ in the direct problem to contraction rates in the indirect problem for $f$. We show that there exists an event $A_n$ with $P_{X,Y}(A_n) \to 1$, such that for any $M_n \to \infty$

$$E_{f_0} \Pi \left[ f \colon \|f - f_0\|_{L_2(\mathcal{T};\mu)} \geq M_n \varepsilon_n^{\mathrm{inv}} \mid X \right] \mathbb{1}_{A_n} \leq C e^{-c\left( n^{\frac{1}{1+2\gamma}} \wedge M_n^2 \right) n\varepsilon_n^2}. \tag{26}$$

The proof follows the lines of Lemma 2.1 of [32]. Let us define

$$\mathcal{S}_n := \left\{ f \in L_2(\mathcal{T};\mu) \colon \sum_{j > k_n} \langle f, e_j \rangle^2 \leq r\rho_n^2 \right\},$$

where the parameters $k_n, r > 0$ and $\rho$ will be specified later, depending on the degree of ill-posedness. Then let us define the modulus of continuity as

$$\delta_n = \sup \left\{ \|f - f_0\|_{L_2(\mathcal{T};\mu)} \colon f \in \mathcal{S}_n, \|\mathcal{A}f - \mathcal{A}f_0\|_{L_2(\mathcal{X};G)} \leq M_n \varepsilon_n \right\} \tag{27}$$

and note that in view of (3.4) from [31]

$$\delta_n \lesssim M_n \kappa_{k_n}^{-1} \varepsilon_n + \rho_n + k_n^{-\beta}. \tag{28}$$

Furthermore, the definition of $\delta_n$ implies that

$$
\begin{aligned}
E_{f_0} \Pi & \left[ f \colon \|f - f_0\|_{L_2(\mathcal{T};\mu)} \geq \delta_n \mid X, Y \right] \mathbb{1}_{A_n} \\
& \leq E_{f_0} \Pi \left[ f \in \mathcal{S}_n \colon \|f - f_0\|_{L_2(\mathcal{T};\mu)} \geq \delta_n \mid X, Y \right] \mathbb{1}_{A_n} + E_{f_0} \Pi \left[ \mathcal{S}_n^c \mid X, Y \right] \mathbb{1}_{A_n} \\
& \leq E_{f_0} \Pi \left[ f \in \mathcal{S}_n \colon \|\mathcal{A}f - \mathcal{A}f_0\|_{L_2(\mathcal{X};G)} \geq M_n \varepsilon_n \mid X, Y \right] \mathbb{1}_{A_n} + E_{f_0} \Pi \left[ \mathcal{S}_n^c \mid X, Y \right] \mathbb{1}_{A_n}.
\end{aligned}
$$

In view of (24) the first term on the right hand side tends to zero for any $A_n \subset D_n$. We show below both in the mildly and severely ill-posed inverse problems, that for appropriate choices of $k_n, \rho_n$ and $r > 0$, we have $\delta_n \lesssim M_n \varepsilon_n^{inv}$ and the second term on the right hand side of the previous display tends to zero.

First we consider the mildly ill-posed problem and set

$$k_n = n^{\frac{1}{1+2\alpha+2p}}, \quad \rho_n = M_n n^{-\frac{\alpha\wedge\beta}{1+2\alpha+2p}}, \quad \varepsilon_n = n^{-\frac{\alpha\wedge\beta+p}{1+2\alpha+2p}}.$$

Then, in view of (28) we have $\delta_n \lesssim M_n \varepsilon_n^{inv}$, hence it remains to show that

$$E_{f_0} \Pi \left[ \mathcal{S}_n^c \mid X, Y \right] \mathbb{1}_{A_n} \lesssim e^{-cM_n^2 n\varepsilon_n^2}. \tag{29}$$

Note that Lemma 5.2 of [32] for $r > 2(1+2\alpha)/\alpha$ (remarking that $\rho_n^2 k_n^{1+2\alpha} = M_n^2 n\varepsilon_n^2 =: n\epsilon_n^2$) provides that

$$\Pi \left[ \mathcal{S}_n^c \right] \leq e^{-Cn(M_n \varepsilon_n)^2}. \tag{30}$$

This together with (25) imply in view of Lemma 4 (with $\epsilon_n = M_n \varepsilon_n$) the inequality (29).

We now turn to the severely ill-posed case and set $k_n = J_{\varepsilon_n} = O(\log^{1/p} n)$ and $\rho_n = M_n \log^{-\beta/p} n$. Since $J_{\varepsilon_n}^\beta e^{cJ_{\varepsilon_n}^p} \asymp \varepsilon_n^{-1}$ it implies $\kappa_{k_n}^{-1} \varepsilon_n \lesssim e^{cJ_{\varepsilon_n}^p} \varepsilon_n \lesssim J_{\varepsilon_n}^\beta \lesssim \log^{-\beta/p} n$, therefore, in view of the arguments above it only remains to show (30). We proceed as in the proof of Lemma 5.2 of [32] and find that, for $Z_j \sim^{iid} N(0,1)$, whenever $t < (2\lambda_j)^{-1}$ for $j > k_n$,

$$\Pi\left[\mathcal{S}_n^c\right] = P\Big(\sum_{j>k_n} \lambda_j Z_j^2 > r\rho_n^2\Big)$$

$$= P\Big(\exp\Big(t\sum_{j>k_n} \lambda_j Z_j^2\Big) > \exp\big(tr\rho_n^2\big)\Big)$$

$$\leq \exp\big(-tr\rho_n^2\big) E \exp\Big(t\sum_{j>k_n} \lambda_j Z_j^2\Big)$$

$$= \exp\big(-tr\rho_n^2\big) \prod_{j>k_n} E \exp\big(t\lambda_j Z_j^2\big)$$

$$= \exp\big(-tr\rho_n^2\big) \prod_{j>k_n} \big(1 - 2t\lambda_i\big)^{-1/2}.$$

Since $\log(1-y) \geq -y/(1-y)$ for $y < 1$,

$$\log\Pi\left[\mathcal{S}_n^c\right] \leq -rt\rho_n^2 + \sum_{j>k_n} \frac{t\lambda_j}{1 - 2t\lambda_j}.$$

Choosing $t = \lambda_{k_n}^{-1}/4$, the second term on the right-hand side above is upper-bounded by a constant. As

$$t\rho_n^2 \asymp M_n^2 \lambda_{k_n}^{-1} \log^{-2\beta/p} n \asymp M_n^2 k_n^\alpha e^{\xi k_n^p} \log^{-2\beta/p} n$$

$$\asymp M_n^2 k_n^\alpha \big(n k_n^{-\alpha}\big)^{\xi/(\xi+2c)} \log^{-2\beta/p} n$$

$$\asymp M_n^2 n^{\xi/(\xi+2c)} (\log n)^{-\frac{2\beta}{p} + \frac{2c\alpha}{p(\xi+2c)}} = M_n^2 n\varepsilon_n^2,$$

the result is proved with $r$ large enough.

### B.4 Step 4: Contraction rate for the VB posterior

Finally, we replace the true posterior by the variational posterior $\Psi^*$ in (26). We can apply Lemma 3 with $\Delta_n = n(M_n\varepsilon_n)^2$ so that, for $M_n \to \infty$,

$$E_{f_0}\Psi^*\left[f: \|f - f_0\|_{L_2(T;\mu)} \geq M_n\varepsilon_n^{\text{inv}}\right] \mathbb{1}_{A_n}$$

$$\leq \frac{2}{\big(n^{\frac{1}{1+2\gamma}} \wedge M_n^2\big) n\varepsilon_n^2}\left(E_{f_0}KL(\Psi^*\|\Pi[\cdot \mid X, Y])\mathbb{1}_{A_n(X,Y)} + Ce^{-\big(n^{\frac{1}{1+2\gamma}} \wedge M_n^2\big)n\varepsilon_n^2/2}\right).$$

Since, $M_n \to \infty$ and $n\varepsilon_n^2 \to \infty$, the conclusion then follows if $E_0 KL(\Psi^*\|\Pi[\cdot \mid X, Y]) \leq Cn\varepsilon_n^2$. According to Lemma 3 in [47] and (19), for any $h \in \mathbb{H}_{\mathcal{A}}$,

$$E_{f_0}KL(\Psi^*\|\Pi[\cdot \mid X, Y]) \leq \sigma^{-2}\big(n\|\mathcal{A}f_0 - h\|_{L_2(\mathcal{X};G)}^2 + \|h\|_{\mathbb{H}_{\mathcal{A}}}^2 E_x \|K_{\mathcal{A}f\mathcal{A}f} - Q_{\mathcal{A}f\mathcal{A}f}\|$$

$$+ E_x Tr\left(K_{\mathcal{A}f\mathcal{A}f} - Q_{\mathcal{A}f\mathcal{A}f}\right)\big).$$

Then in view of Lemma 1, for $n$ large enough, there exists $h \in \mathbb{H}_{\mathcal{A}}$ such that $\|\mathcal{A}f_0 - h\|_{L_2(\mathcal{X};G)} \leq \varepsilon_n$ and $\|h\|_{\mathbb{H}_{\mathcal{A}}} \leq n\varepsilon_n^2$. Hence the claimed upper bound follows from the assumptions on the trace and spectral norm term.

## B.5 Technical lemmas

**Lemma 1** (RKHS approximation for random series priors). *Let $f_0 \in \bar{H}^\beta$, $\beta > 0$, and consider the centered GP prior $\Pi_{\mathcal{A}}$ on $\mathcal{A}f$ given in (15). Then*

$$\inf_{h \in \mathbb{H}_{\mathcal{A}} : \|h - \mathcal{A}f_0\|_{L_2(\mathcal{X};G)} \leq \epsilon} \|h\|_{\mathbb{H}_{\mathcal{A}}}^2 \lesssim \begin{cases} \epsilon^{-\frac{2\alpha - 2\beta + 1}{\beta + p}} & \text{if } \kappa_j \asymp j^{-p}, \lambda_j \asymp j^{-1-2\alpha} \text{ for } \alpha > 0, p \geq 0, \ \beta \leq 2\alpha + 1 \\[2mm] J_\epsilon^{\alpha - 2\beta} e^{\xi J_\epsilon^p} & \text{if } \kappa_j \asymp e^{-cj^p}, \lambda_j \asymp j^{-\alpha} e^{-\xi j^p}, \text{ for } \alpha \geq 0, \xi > 0 \text{ or} \\ & \quad \xi = 0, \alpha \geq 2\beta, \text{ and } p \geq 1, \end{cases}$$

*where $J_\epsilon$ is the smallest integer such that $\max_{j \geq J_\epsilon}(\kappa_j j^{-\beta})\|f_0\|_\beta \leq \epsilon$.*

*Proof.* For simplicity let us denote by $w = \mathcal{A}f_0$ and note that for any $J \in \mathbb{N}$, the function $w^J(x) = \sum_{j=1}^J w_j g_j(x) \in \mathbb{H}_{\mathcal{A}}$, with $w_j = \langle w, g_j \rangle_{L_2(\mathcal{X};G)}$. Then in view of (16) and using the notation $f_{0,j} = \langle f_0, e_j \rangle_{L_2(\mathcal{T};\mu)}$,

$$\|w^J\|_{\mathbb{H}_{\mathcal{A}}}^2 = \sum_{j=1}^J \kappa_j^{-2} \lambda_j^{-1} w_j^2 = \sum_{j=1}^J j^{-2\beta} \lambda_j^{-1} f_{0,j}^2 j^{2\beta} \leq \max_{1 \leq j \leq J}(j^{-2\beta} \lambda_j^{-1}) \|f_0\|_\beta^2,$$

$$\|w^J - w\|_{L_2(\mathcal{X};G)}^2 = \sum_{j=J+1}^\infty w_j^2 = \sum_{j=J+1}^\infty \kappa_j^2 j^{-2\beta} f_{0,j}^2 j^{2\beta} \leq \max_{j \geq J}(\kappa_j^2 j^{-2\beta}) \|f_0\|_\beta^2.$$

Then, in the mildly ill-posed inverse problem (with $\kappa_j \asymp j^{-p}, \lambda_j \asymp j^{-1-2\alpha}$), the smallest $J_\epsilon \in \mathbb{N}$ such that $\max_{j \geq J_\epsilon}(\kappa_j j^{-\beta})\|f_0\|_\beta \leq \epsilon$ satisfies that $J_\epsilon \asymp (\|f_0\|_\beta/\epsilon)^{1/(\beta+p)}$, resulting in $\|w^{J_\epsilon}\|_{\mathbb{H}_{\mathcal{A}}}^2 \lesssim \epsilon^{-\frac{2\alpha - 2\beta + 1}{\beta + p}}$ and proving the first statement. In the severely ill-posed case (with $\kappa_j \asymp e^{-cj^p}, \lambda_j \asymp j^{-\alpha} e^{-\xi j^p}$) the smallest $J_\epsilon \in \mathbb{N}$ such that $\max_{j \geq J_\epsilon}(\kappa_j j^{-\beta})\|f_0\|_\beta \leq \epsilon$ implies that $\|w^{J_\epsilon} - w\|_{L_2(\mathcal{X};G)}^2 \lesssim J_\epsilon^{\alpha - 2\beta} e^{\xi J_\epsilon^p}$ $\qquad\square$

**Lemma 2** (Small ball probability for random series priors). *Consider the centered GP prior $\Pi_{\mathcal{A}}$ on $\mathcal{A}f$ given in (15). Then there exists $C > 0$ depending on $\alpha, p, c, \xi$ such that for any $\epsilon > 0$ small enough*

$$-\log \Pi_{\mathcal{A}}\left(w : \|w\|_{L_2(\mathcal{X};G)} < \epsilon\right) \leq C \begin{cases} \epsilon^{-1/(\alpha + p)} & \text{if } \kappa_j \asymp j^{-p}, \lambda_j \asymp j^{-1-2\alpha} \text{ for } \alpha > 0, p \geq 0, \\[2mm] \log^{(p+1)/p} \frac{1}{\epsilon} & \text{if } \kappa_j \asymp e^{-cj^p}, \lambda_j \asymp j^{-\alpha} e^{-\xi j^p}, \\ & \quad \text{for } \alpha \geq 0, \xi > 0 \text{ or } \xi = 0, \alpha \geq 2\beta, \text{ and } p \geq 1. \end{cases}$$

*Proof.* The first case (polynomial decay) was derived in Lemma 11.47 from [23]. In the second case, for $J \geq 1$ and $Z_j \sim^{iid} N(0,1)$,

$$\Pi_{\mathcal{A}}\left(w : \|w\|_{L_2(\mathcal{X};G)} < \epsilon\right) \geq P\left(\sum_{j \leq J} \lambda_j \kappa_j^2 Z_j^2 < \epsilon^2/2\right) P\left(\sum_{j > J} \lambda_j \kappa_j^2 Z_j^2 < \epsilon^2/2\right).$$

Note that the likelihood ratio of centered Gaussians with standard deviations $\sigma \geq \tau$ satisfy $\psi_\sigma/\psi_\tau(x) \geq \tau/\sigma$ uniformly on $x \in \mathbb{R}$. Therefore, the first term on the rhs of the preceding display is bounded from below by

$$P\left(\sum_{j \leq J} j^{-\alpha} e^{-(\xi + 2c)j^p} Z_j^2 < c\epsilon^2\right) \geq$$

$$e^{(c+\xi/2)\left(\sum_{j=1}^J j^p - J^{p+1}\right)} \prod_{j=1}^J \left(\frac{j}{J}\right)^{\alpha/2} P\left(\sum_{j \leq J} J^{-\alpha} e^{-(\xi + 2c)J^p} Z_j^2 < c\epsilon^2\right).$$

The logarithm of the leading factor is equivalent to $-\frac{p}{p+1}(c + \xi/2)J^{p+1}$ as $J \to \infty$. The second is lower bounded by $(J!/J^J)^{\alpha/2} \geq e^{-J\alpha/2}$. By the central limit theorem, the probability in the last factor is greater than $1/2$ as $J \to \infty$ as long as $J^{\alpha-1} e^{(\xi+2c)J^p} \epsilon^2 c \geq 2$. Also, by Markov's inequality,

$$P\left(\sum_{j > J} \lambda_j \kappa_j^2 Z_j^2 < \epsilon^2/2\right) \geq 1 - 2\epsilon^{-2} \sum_{j > J} E(Z_j^2 \lambda_j \kappa_j^2) \geq 1 - c_1 \epsilon^{-2} \sum_{j > J} j^{-\alpha} e^{-(\xi + 2c)j^p}.$$

Since the above sum is smaller than $c_2\epsilon^{-2}J^{-\alpha}e^{-(\xi+2c)J^p}$ (following form the assumption $p \geq 1$ and the sum of geometric series), the above probability is greater than $1/2$ whenever $J^\alpha e^{(\xi+2c)J^p} \geq 2c_2\epsilon^{-2}$. Therefore, as long as $J^{\alpha-1}e^{(\xi+2c)J^p}\epsilon^2 \geq (2/c) \vee (2c_2)$,

$$-\log \Pi_{\mathcal{A}}\left(w : \|w\|_{L_2(\mathcal{X};G)} < \epsilon\right) \lesssim J^{p+1}.$$

The above conditions are satisfied for $J \asymp \log^{1/p}\epsilon^{-1}$, concluding the proof of the lemma. $\qquad\square$

**Lemma 3** (Theorem 5 of [58]). *Let $C_n$ be a measurable subset of the parameter space $L_2(\mathcal{T};\mu)$, $A_n$ be an event and $Q$ a distribution on $L_2(\mathcal{T};\mu)$. If there exists $C > 0$ and $\Delta_n \to \infty$ such that*

$$E_{f_0}\Pi\left[C_n^c \mid X, Y\right]\mathbb{1}_{A_n} \leq Ce^{-\Delta_n},$$

*then*

$$E_{f_0}Q\left(C_n^c\right)\mathbb{1}_{A_n} \leq \frac{2}{\Delta_n}\left[E_{f_0}KL\big(Q\|\Pi[\cdot \mid X, Y]\big) + Ce^{-\Delta_n/2}\right].$$

**Lemma 4.** *Let $\mathcal{S}_n \subset L_2(\mathcal{T};\mu)$ be a measurable event such that for some $\epsilon_n \to 0$, $n\epsilon_n^2 \to \infty$, and $C > 1$ large enough,*

$$\frac{\Pi[\mathcal{S}_n]}{\Pi[f: \|\mathcal{A}f - \mathcal{A}f_0\|_{L_2(\mathcal{X};G)} \leq \epsilon_n]} \leq e^{-Cn\epsilon_n^2}.$$

*Then there exists an event $A_n \subset \mathcal{X}^n$, with $P_{f_0}(A_n) \to 1$, and $C' > C/2$ such that*

$$E_{f_0}\Pi\left[\mathcal{S}_n \mid X, Y\right]\mathbb{1}_{A_n} \lesssim e^{-C'n\epsilon_n^2}.$$

*Proof.* For $KL(f_0\|f) = P_{f_0}\log\left(dP_{f_0}/dP_f\right)$ and $V(f_0\|f) = P_{f_0}|\log\left(dP_{f_0}/dP_f\right)|^2$, in the random design regression model, in view of Lemma 2.7 of [23], the neighbourhood

$$B_2(f_0;\epsilon_n) := \left(f: KL(f_0\|f) \leq n\epsilon_n^2,\ V(f_0\|f) \leq n\epsilon_n^2\right)$$

contains the ball $\left\{f: \|\mathcal{A}f - \mathcal{A}f_0\|_{L_2(\mathcal{X};G)} \leq \epsilon_n\right\}$. Therefore,

$$\Pi\big[f: \|\mathcal{A}f - \mathcal{A}f_0\|_{L_2(\mathcal{X};G)} \leq \epsilon_n\big] \leq \Pi\left[B_2(f_0;\epsilon_n)\right].$$

By Lemma 8.10 in [23], for any $c > 1$, there exists an event $A_n^c$ of vanishing mass such that on $A_n$

$$\int dP_f/dP_{f_0}(X, Y)\Pi(df) \geq \Pi\left[B_2(f_0;\epsilon_n)\right]e^{-cn\epsilon_n^2}.$$

Let us define $B_n = A_n \cap \{\psi = 0\}$ for any $\psi: (\mathcal{X} \times \mathbb{R})^n \mapsto \{0, 1\}$ such that $E_{f_0}\psi \to 0$, implying $P(B_n^c) = o(1)$. Then, taking $c < C$,

$$
\begin{aligned}
E_{f_0}\Pi\left[\mathcal{S}_n \mid X, Y\right]\mathbb{1}_{B_n} &= E_{f_0}\frac{\int_{\mathcal{S}_n} dP_f/dP_{f_0}(X, Y)\left(1 - \psi\right)(X, Y)d\Pi(f)}{\int dP_f/dP_{f_0}(X, Y)d\Pi(f)}\mathbb{1}_{B_n}\\
&\lesssim e^{cn\epsilon_n^2}\frac{\int_{\mathcal{S}_n} E_{f_0}dP_f/dP_{f_0}\left(1 - \psi\right)d\Pi(f)}{\Pi[B_2(f_0;\epsilon_n)]}\\
&\lesssim e^{cn\epsilon_n^2}\frac{\int_{\mathcal{S}_n} E_f\left(1 - \psi\right)d\Pi(f)}{\Pi[f: \|\mathcal{A}f - \mathcal{A}f_0\|_{L_2(\mathcal{X};G)} \leq \epsilon_n]}\\
&\lesssim e^{cn\epsilon_n^2}\frac{\Pi[\mathcal{S}_n]}{\Pi[f: \|\mathcal{A}f - \mathcal{A}f_0\|_{L_2(\mathcal{X};G)} \leq \epsilon_n]} \lesssim e^{-C'n\varepsilon_n^2}.
\end{aligned}
$$

$\qquad\square$

**Lemma 5.** *Assume that $\|g_j\|_\infty \lesssim j^\gamma$ and that $\alpha + p > 1 + 2\gamma$ in case of the mildly ill-posed inverse problem. Then, there exists an event $B_n \subset \mathcal{X}^n$ with $P_X(B_n^c) = o(1)$, $C > 0$ and a measurable subset $\mathcal{G}_n \subset L_2(\mathcal{X};G)$ with $\Pi(f: \mathcal{A}f \in \mathcal{G}_n^c) = o(e^{-n^{\frac{1}{1+2\gamma}}n\varepsilon_n^2})$ satisfying*

$$\|w\|_{L_2(\mathcal{X};G)}^2 \leq C(\|w\|_{L_2(\mathcal{X};P_n)}^2 + \varepsilon_n^2) \tag{31}$$

*and*

$$\|w\|_{L_2(\mathcal{X};P_n)}^2 \leq C(\|w\|_{L_2(\mathcal{X};G)}^2 + \varepsilon_n^2) \tag{32}$$

*for any $X \in B_n$ and $w \in \mathcal{G}_n$.*

*Proof.* Let us take $k = n^{\frac{1}{1+2\gamma}} / \log^{2/(1+2\gamma)} n$ and define the sieve

$$\mathcal{F}_n = \{f \in L_2(\mathcal{T}; \mu) : \mathcal{A}f \in \mathcal{G}_n\}, \text{ with } \mathcal{G}_n = \{w \in L_2(\mathcal{X}; G) : \|w^{\perp k}\|_\infty \leq \varepsilon_n\}, \qquad (33)$$

where $w^{\perp k}(x) = \sum_{j=k+1}^\infty w_j g_j(x)$. Similarly we will denote by $w^k(x) = \sum_{j=1}^k w_j g_j(x)$. Next we show that $\Pi(\mathcal{F}_n^c) = o(e^{-n^{\frac{1}{1+2\gamma}} n \varepsilon_n^2})$.

First note that the assumption $\|g_j\|_\infty \lesssim j^\gamma$ implies that $\|w^{\perp k}\|_\infty \leq C \sum_{j=k+1}^\infty j^\gamma |w_j|$. Under the prior $\Pi$ on $f$, we have $f_j = \langle f, g_j \rangle_{L_2(\mathcal{X}; G)} \stackrel{d}{=} \lambda_j^{1/2} Z_j$ with $Z_j \sim^{iid} N(0, 1)$, therefore

$$\Pi(f : \|\mathcal{A}f^{\perp k}\|_\infty > \varepsilon_n) \leq P\Big(C \sum_{j=k+1}^\infty \kappa_j \lambda_j^{1/2} j^\gamma |Z_j| > \varepsilon_n\Big) = o(e^{-n^{\frac{1}{1+2\gamma}} n \varepsilon_n^2}),$$

where the last equation follows from Lemma 7 and $\alpha + p > 1 + 2\gamma$. The above two displays together imply that $\Pi(f : \mathcal{A}f \in \mathcal{G}_n^c) \lesssim e^{-C^o n \varepsilon_n^2}$.

It remains to show that for $w \in \mathcal{G}_n$ there exists an event $B_n$ with $P_X(B_n) \to 1$, such that for $X \in B_n$ the inequalities (31) and (32) hold. This follows from the fact that for $w \in \mathcal{G}_n$,

$$\|w^{\perp k}\|_{L_2(\mathcal{X}; G)} \vee \|w^{\perp k}\|_{L_2(\mathcal{X}; P_n)} \leq \|w^{\perp k}\|_\infty \leq \varepsilon_n.$$

This inequality also allows to write, under the event of Lemma 6 which we note $B_n$, that

$$\|w\|_{L_2(\mathcal{X}; P_n)} \leq \|w^k\|_{L_2(\mathcal{X}; P_n)} + \|w^{\perp k}\|_{L_2(\mathcal{X}; P_n)} \lesssim \|w^k\|_{L_2(\mathcal{X}; G)} + \varepsilon_n \leq \|w\|_{L_2(\mathcal{X}; G)} + \varepsilon_n,$$

proving (31) (a similar argument proves (32)).

$\square$

**Lemma 6.** *For $k = n^{\frac{1}{1+2\gamma}} / \log^{2/(1+2\gamma)} n$, $\gamma \geq 0$, there exists a constant $C_0 > 1$ such that, with $P_X$-probability tending to one,*

$$C_0^{-1} \|w^k\|_{L_2(\mathcal{X}; G)} \leq \|w^k\|_{L_2(\mathcal{X}; P_n)} \leq C_0 \|w^k\|_{L_2(\mathcal{X}; G)},$$

*for any $w \in L_2(\mathcal{X}; G)$, and where $w^k(x) = \sum_{j=1}^k w_j g_j(x)$ is the orthogonal projection on the $k$ first elements of an orthonormal basis $(g_j)_{j \in \mathbb{N}}$ satisfying $\|g_j\|_\infty \lesssim j^\gamma$.*

*Proof.* First we introduce $\Sigma_{n,k} = n^{-1} \boldsymbol{G}_{n,k}^T \boldsymbol{G}_{n,k}$ with $\boldsymbol{G}_{n,k} = (\boldsymbol{g}(X_1), ...., \boldsymbol{g}(X_n))^T \in \mathbb{R}^{n \times k}$, with $\boldsymbol{g}(X_1) = (g_1(X_1), ..., g_k(X_1))^T$. Note that $E_{\mathbb{X}} \Sigma_{n,k} = I_k$ as the eigenbasis $(g_j)_{j \in \mathbb{N}}$ is orthonormal w.r.t. the design distribution $G$. Then by the modified version of Rudelson's inequality [62] we get that

$$E_{\mathbb{X}} \|\Sigma_{n,k} - I_k\|_2 \leq C \sqrt{\frac{\log k}{n}} E_{\mathbb{X}}(\|\boldsymbol{g}(X_1)\|_2^{\log n})^{1/\log n}.$$

Note that by the boundedness assumption $\sum_{j=1}^k g_j(x)^2 \leq C k^{1+2\gamma}$, $x \in \mathcal{X}$, so that the right hand side of the preceding display is bounded from above by constant times $\sqrt{k^{1+2\gamma} \log(k)/n} = o(1)$. Therefore, noting $\boldsymbol{w} = (w_1, ..., w_k)$ for $w \in L_2(\mathcal{X}; G)$,

$$
\begin{aligned}
\sup_{w \in L_2(\mathcal{X}; G)} \frac{\Big| \|w^k\|_{L_2(\mathcal{X}; G)}^2 - \|w^k\|_{L_2(\mathcal{X}; P_n)}^2 \Big|}{\|w^k\|_{L_2(\mathcal{X}; G)}^2} &= \sup_{w \in L_2(\mathcal{X}; G)} \Big| \boldsymbol{w}^T (I_k - \Sigma_{n,k}) \boldsymbol{w} \Big| / \|w^k\|_{L_2(\mathcal{X}; G)}^2 \\
&\leq \sup_{w \in L_2(\mathcal{X}; G)} \|I_k - \Sigma_{n,k}\|_2 \|\boldsymbol{w}\|_2^2 / \|w^k\|_{L_2(\mathcal{X}; G)}^2 \\
&= o_{P_{\mathbb{X}}}(1).
\end{aligned}
$$

Then, on an event $A_n(\mathbb{X})$ with $P_{\mathbb{X}}(A_n(\mathbb{X}))$ tending to one, for all $w \in \mathcal{G}_n$

$$\|w^k\|_{L_2(\mathcal{X}; G)}^2 / 2 \leq \|w^k\|_{L_2(\mathcal{X}; P_n)}^2 \leq 2 \|w^k\|_{L_2(\mathcal{X}; G)}^2, \qquad (34)$$

for any $w$, verifying the statement. $\square$

**Lemma 7.** *Assume that $\nu_j \le Cj^{-3/2-\delta}$, $\delta > \gamma \ge 0$ and that $n\varepsilon_n^2 \to \infty$. Then, for $Z_j$ independent standard normal random variables and any $C' > 0$,*

$$P\left(\sum_{j=n^{\frac{1}{1+2\gamma}}/\log^{2/(1+2\gamma)} n}^{\infty} \nu_j |Z_j| \ge C'\varepsilon_n\right) = o(e^{-n^{\frac{1}{1+2\gamma}} n\varepsilon_n^2}).$$

*Proof.* Let us introduce the notation $k = n^{\frac{1}{1+2\gamma}}/\log^{2/(1+2\gamma)} n$ and note that, for any $C_1 > 0$, there exist positive constants $C_2, C_3$ such that

$$
\begin{aligned}
P\left(\sum_{j=k}^{\infty} \nu_j |Z_j| \ge C_1\varepsilon_n\right) &\le \sum_{i=1}^{\infty} P\left(\sum_{j=ik}^{(i+1)k-1} \nu_j |Z_j| \ge C_2\left(i\left(1+\log^2 i\right)\right)^{-1}\varepsilon_n\right) \\
&\le \sum_{i=1}^{\infty} P\left(\sum_{j=ik}^{(i+1)k-1} Ci^{-3/2-\delta}k^{-3/2-\delta}|Z_j| \ge C_2\left(i(1+\log^2 i)\right)^{-1}\varepsilon_n\right) \\
&\le \sum_{i=1}^{\infty} P\left(\sum_{j=ik}^{(i+1)k-1} |Z_j| \ge C_3 k^{3/2+\delta}i^{1/2}\varepsilon_n\right). \quad\quad (35)
\end{aligned}
$$

We show below that

$$P\left(\sum_{j=ik}^{(i+1)k-1} |Z_j| \ge C_3 k^{3/2+\delta}i^{1/2}\varepsilon_n\right) \le 2^k e^{-cik^{2+2\delta}\varepsilon_n^2}. \quad\quad (36)$$

which in turn implies (together with $n\varepsilon_n^2 \to \infty$ and $\delta > \gamma \ge 0$) that the rhs of (35) is further bounded by

$$\sum_{i=1}^{\infty} 2^k e^{-c_1 ik^{2+2\delta}\varepsilon_n^2} \lesssim 2^k e^{-c_1 k^{2+2\delta}\varepsilon_n^2} = o(e^{-n^{\frac{1}{1+2\gamma}} n\varepsilon_n^2}).$$

It remained to prove (36). For convenience, let us introduce the notation $c_{i,k} = C_3 i^{1/2}k^{3/2+\delta}$. Following the proof of Chernoff's inequality and recalling that the characteristic function of the absolute value of the standard normal distribution satisfies that $Ee^{t|Z|} \le 2e^{t^2/2}$, we get for $\gamma = c_{i,k}\varepsilon_n/k$ that

$$
\begin{aligned}
P\left(\sum_{j=ik}^{(i+1)k-1} |Z_j| \ge c_{i,k}\varepsilon_n\right) &= P\left(e^{\gamma \sum_{j=ik}^{(i+1)k-1} |Z_j|} \ge e^{\gamma c_{i,k}\varepsilon_n}\right) \\
&\le e^{-\gamma c_{i,k}\varepsilon_n} Ee^{\gamma \sum_{j=ik}^{(i+1)k-1} |Z_j|} \le e^{-\gamma c_{i,k}\varepsilon_n} \prod_{j=ik}^{(i+1)k-1} 2e^{\gamma^2/2} \\
&= 2^k e^{k\gamma^2/2 - \gamma c_{i,k}\varepsilon_n} = 2^k e^{-c_{i,k}^2\varepsilon_n^2/(2k)}.
\end{aligned}
$$

$\square$

**Lemma 8.** *Let $f_0 \in \bar{H}^\beta$, for some $\beta > 0$, $J \in \mathbb{N}$ and assume that $\|g_j\|_\infty \le Cj^\gamma$ for some $\gamma \ge 0$. Furthermore, in case of the mildly ill-posed inverse problem assume that $p + \beta - \gamma > 1$. Then $P_X$-almost surely*

$$\|\mathcal{A}f_0^{\perp J}\|_{L^2(\mathcal{X}, P_n)} \lesssim q(J),$$

*where $q(J) = J^{-p-\beta+\gamma+1}$ in the mildly and $q(J) = J^{\gamma-\beta}e^{-cJ^p}$ in the severely ill-posed inverse problem. Furthermore, for any $J \le k = n^{\frac{1}{1+2\gamma}}/\log^{2/(1+2\gamma)} n$, with $P_X$-probability tending to one*

$$\|\mathcal{A}f_0^{\perp J}\|_{L^2(\mathcal{X}, P_n)} \lesssim \|\mathcal{A}f_0^{\perp J}\|_{L^2(\mathcal{X}, G)} + q\left(n^{\frac{1}{1+2\gamma}}/\log^{2/(1+2\gamma)} n\right).$$

*Proof.* We start with the first assertion. In view of $|f_{0,j}| \leq j^{-\beta}\|f_0\|_\beta$ and triangle inequality one can observe that

$$\|\mathcal{A}f_0^{\perp J}\|_{L^2(\mathcal{X},P_n)} \leq \|\mathcal{A}f_0^{\perp J}\|_\infty \leq \sum_{j=J+1}^\infty \kappa_j |f_{0,j}| \|g_j\|_\infty \lesssim \sum_{j=J+1}^\infty \kappa_j j^{\gamma-\beta}.$$

Then for the mildly ill-posed inverse problem (with $\kappa_j \asymp j^{-p}$) the rhs of the preceding display is further bounded by a multiple of $J^{-p-\beta+\gamma+1}$, while in the severely ill-posed inverse problem (with $\kappa_j \asymp e^{-cj^p}$) it is bounded from above by a multiple of $J^{\gamma-\beta}e^{-cJ^p}$ since $p \geq 1$.

For the second assertion of the lemma, note that for $J \leq k = n^{\frac{1}{1+2\gamma}}/\log^{2/(1+2\gamma)} n$, by triangle inequality

$$\|\mathcal{A}f_0^{\perp J}\|_{L^2(\mathcal{X},P_n)} \leq \|\mathcal{A}f_0^{\perp J} - \mathcal{A}f_0^{\perp k}\|_{L^2(\mathcal{X},P_n)} + \|\mathcal{A}f_0^{\perp k}\|_{L^2(\mathcal{X},P_n)}.$$

The first term, in view of Lemma 6, is bounded by a multiple of $\|\mathcal{A}f_0^{\perp J} - \mathcal{A}f_0^{\perp k}\|_{L^2(\mathcal{X},G)} \leq \|\mathcal{A}f_0^{\perp J}\|_{L^2(\mathcal{X},G)}$ with $P_X$-probability tending to one by, while the second term is bounded by a multiple of $q\left(n^{\frac{1}{1+2\gamma}}/\log^{2/(1+2\gamma)} n\right)$ following from the first statement of the lemma. $\square$

## C  Proof of Corollary 1

For the first choice (9) in the mildy ill-posed case, Lemma 4 of [47] combined with the polynomial decay of the eigenvalues $\lambda_i \kappa_i^2$ of the process with kernel (15) gives that

$$E_X \|K_{\mathcal{A}f\mathcal{A}f} - Q_{\mathcal{A}f\mathcal{A}f}\| \lesssim nm^{-1-2(\alpha+p)},$$
$$E_X Tr\left(K_{\mathcal{A}f\mathcal{A}f} - Q_{\mathcal{A}f\mathcal{A}f}\right) \lesssim nm^{-2(\alpha+p)}.$$

In view of Theorem 1, we set

$$m = m_n = n^{\frac{1}{1+2p+2\alpha}}$$

to translate posterior contraction rates into variational ones. For the second case (10), since $\alpha + p > 1$ and $\sup_j \sup_x |g_j(x)| < \infty$ under our assumptions, the bounds come from Lemma 5 of [47] and are

$$E_X \|K_{\mathcal{A}f\mathcal{A}f} - Q_{\mathcal{A}f\mathcal{A}f}\| \lesssim 1 + nm^{-1-2(\alpha+p)} + n^{\frac{1}{2(\alpha+p)}} m^{-2(\alpha+p)} \log n,$$
$$E_X Tr\left(K_{\mathcal{A}f\mathcal{A}f} - Q_{\mathcal{A}f\mathcal{A}f}\right) \lesssim nm^{-2(\alpha+p)}.$$

Then, $m = m_n$ as above is sufficient as well.

Finishing with the severely ill-posed problem, we have for both choices of inducing variables

$$E_X \|K_{\mathcal{A}f\mathcal{A}f} - Q_{\mathcal{A}f\mathcal{A}f}\| \leq E_X Tr\left(K_{\mathcal{A}f\mathcal{A}f} - Q_{\mathcal{A}f\mathcal{A}f}\right) \leq n \sum_{j>m} \lambda_j \lesssim nm^{-\alpha} e^{-(\xi+2c)m^p},$$

where the second inequality comes from Proposition 2 in [66]. Then, $m = m_n = ((\xi + 2c)\log n)^{1/p}$ is sufficient.

*Funding.* Co-funded by the European Union (ERC, BigBayesUQ, project number: 101041064). Views and opinions expressed are however those of the author(s) only and do not necessarily reflect those of the European Union or the European Research Council. Neither the European Union nor the granting authority can be held responsible for them.