# OpenReview forum: "Variational Gaussian processes for linear inverse problems"
_NeurIPS.cc/2023/Conference — NeurIPS 2023 poster_

### Official Review · Reviewer_Sbdt · 2023-06-30

**Soundness:** 3 good
**Presentation:** 2 fair
**Contribution:** 3 good
**Rating:** 5
**Confidence:** 3

**Summary:**

This paper analyzes posterior contraction rates for the inducing variables variational Bayes approximation for Gaussian process. In particular, mildly and ill-posed inverse problem settings, which are characterized by profiles of eigenvalues of measurement operators, are considered. For the both cases, the posterior contract rates are assessed theoretically. The usefulness of the results are shown for applications to three example problems.

**Strengths:**

The posterior contract rates are assessed analytically for the inducing variable variational Bayes approximation of Gaussian process. This is useful to determine the optimal number of inducing variables in the approximation.

**Weaknesses:**

The analysis relies on the assumption that the functional space where the target function belongs is known. However, there are many practical situations where such an assumption does not hold. The analysis is not applicable for such situations.

**Questions:**

What happens if the target function does not belong to the assumed functional space?

**Limitations:**

Yes.

---

> ### Author Rebuttal · Authors · 2023-08-09
>
> We thank the reviewer for the summary and the question on adaptivity, which is of high importance in practice.
>  As it is common in the literature, our first article focuses on the non-adaptive case when the function space
> is known and derived guarantees in this setting. The next, natural, follow up question is to address adaptation
> to the unknown function class. Please note that for the variational Bayes procedure adaptation to unknown function classes
>  has been considered only for specific examples in the literature focusing mainly on mean-field VB methods. These results
>  can not be directly applied to the inducing variable Gaussian Process  framework (where the variational class is not mean-field)
>  even in the direct case, let alone in the more complicated ill-posed inverse problem setting. Since this is a very challenging
>  question going beyond the scope of our already quite technical and long paper, we have left it to future work.

---

> > ### Comment · Reviewer_Sbdt · 2023-08-13
> >
> > Thank you for the explanation. I will keep the evaluation as it is.

---

### Official Review · Reviewer_KBMK · 2023-07-03

**Soundness:** 3 good
**Presentation:** 3 good
**Contribution:** 2 fair
**Rating:** 6
**Confidence:** 2

**Summary:**

This paper presents a theoretical analysis of the posterior contraction rates for the variational posterior in the context of linear inverse problems. Specifically, the variational posterior is obtained using the sparse Gaussian processes approach, with inducing variables learned through KL divergence minimization. The authors investigate three types of operators: the Volterra operator, the heat equation, and the Radon transform, each exhibiting varying degrees of ill-posedness. The numerical analysis focuses on synthetic toy problems, and the authors demonstrate that in this case, even a small number of inducing points can yield highly accurate results, achieving significant speed-up.

**Strengths:**

[Originality] I am not up-to-date with theoretical results with frequentist-Bayes analysis, and how far the presented results are based on or deviates from the monograph Nickl, Richard, 2023.

Nickl, Richard. "Bayesian non-linear statistical inverse problems." (2023).

[Quality] The theoretical results seems sound.

[Clarity] The paper is well-written; however, the use of jargon limits its accessibility to a narrower audience.

[Significance] The inducing-point of Gaussian processes is widely used, so the theoretical results concerned with it should be of interest to the community working on inverse problems.

**Weaknesses:**

[Quality] The experimental evaluation is limited to a synthetic toy example only.

[Significance] The authors motivate the variational Bayes approach for computational reasons. However, recent developments of scalable GP methods can reduce the complexity of exact GP from O(n^3) to O(n^2) and effectively leverage the modern computing hardware such as GPU (Gardner, Jacob, et al.  2018). In this regard, one can afford many more inducing points in practice.

Gardner, Jacob, et al. "Gpytorch: Blackbox matrix-matrix gaussian process inference with gpu acceleration." *Advances in neural information processing systems* 31 (2018).

The paper is a theory paper, but its practical significance appears to be lacking. It would greatly enhance the paper's value if the authors could at least mention some real-world applications in which the studied inverse problem commonly arises. For instance, discussing the application of the Radon transform in computed tomography (CT) could serve as a relevant and illustrative example.

**Questions:**

Figure 1 shows that using m=3 is insufficient, while m=6 yields highly accurate results for the studied case, while this is a single case not quite informative. I am wondering whether it is possible to establish a "phase-transition curve" that can inform practitioners about the required number of inducing points for various types of inverse problems and different numbers of measurements.

If the results are not understood correctly, it might leads to poor choice on the number of inducing points,  and over-or-under estimate of the posterior uncertainty. This could be harmful in mission-critical applications such as medical imaging. How to prevent this happen?

Could you please further explain why the posterior uncertainty being over-estimated while in traditional variational approximation the uncertainty usually got underestimated? How well is the uncertainty calibrated?

**Limitations:**

The authors do have discussed to what extend the presented results can be extended to nonlinear cases and other types of inverse problems.

---

> ### Author Rebuttal · Authors · 2023-08-09
>
> We thank the referee for his/her interest in the paper and the constructive comments and suggestions.
>
> ## Theoretical literature on frequentist-Bayes analysis
>
> The monograph from Richard Nickl is multiscale analysis applied to non-linear inverse problems. This is a completely differnt techniques basically deriving simultaneous semi-parametric Bernstein-von Mises theorems leading to a Bernstein-von Mises type of result on a weak function space. For the investigated linear-inverse problems it is substantially simpler to follow our strategy deriving contraction rates in the direct problem, then use modulus of continuity and connect the true posterior with its variational approximation.
>
>
> ## Experiments
>
> While our numerical analsys indeed considers only synthetic datasets, we argue that the selected examples are highly relevant and often form the baselines
>  in the literature. For example, the heat equation is often consdiered as the starting example in the PDE literature and for instance the Black-Scholes PDE
>  can be converted to the heat equation as well. In addition, the Radon transform is a mathematical technique with various applications, particularly in the
> field of medical imaging and image analysis. It is used to analyze and transform data from the spatial domain to the Radon domain, which provides a different
>  perspective on the data that can be useful for specific tasks. Inverting the Radon transform has found a lot of applications where lower-dimensional integrals
>  of the inside of an object are more readily available than the object of interest itself. For instance, in Computed Tomography (CT) Imaging, X-ray measurements
>  are taken from different angles around a patient, and the Radon transform is used to reconstruct a cross-sectional image (slice) of the patient's body.
>  This helps doctors visualize internal structures and diagnose various medical conditions. Medical Single Photon Emission Computed Tomography (SPECT) is a nuclear
>  medicine imaging technique that uses gamma-ray detectors to generate 3D images of the distribution of radioactive tracers within a patient's body, applying the
>  Radon transform in the image reconstruction process. [*Barrett*, 1984,  *Natterer and Wübbeling*, 2001, *Ramlau and Scherzer*, 2019, *Ambartsoumian and Quinto*, 2020].
> Further applications include Seismic Imaging, Geophysical imaging, Radar and Sonar Imaging, Particle Tracking and Material Science and Crystallography, which we
> all discuss in details (with references) in the supplement of our manuscript.
>
> ## Recent developments on scalable GPs
>
> We sincerely thank the reviewer for bringing the paper on GPytorch to our knowledge. While their method improves the scalability of both exact and sparse approximate
>  computations of posterior GPs, this should not encourage us to include many more inducing variables in our approach. Indeed, one of the main results of our paper is
> a sufficient lower bound on the number of variables we need.  Adding more won't improve substantially the quality of the inference as demonstrated by our theory and numerical analysis.
> Nevertheless, the combination of scalable approaches will allow us to tackle larger data sets and/or less regular models more efficiently. Please also note that our approach
> provides substantial speed up in case of severely ill-posed inverse problems (where almost linear computational complexity applies) and for highly irregular functions.
>
> ## Phase-transition curve
>
> We thank the reviewer for this nice idea. In our general response we have addressed this comment in details.
>
>
> ## Uncertainty quantification
>
> We thank the reviewer for this really interesting question. The observation that usual variational methods tend to under-estimate the posterior uncertainty
>  is based on the study of mean-field variational method or multimodal posterior distributions estimated by a unimodal one, see e.g. "Bishop: Pattern recognition and machine learning". However,
> our case does not fall into these categories. The inducing variable GP approximation has two components. Conditioned on the inducing points its law is coming
>  from the GP prior. Then the inducing variables are fitted to best approximate the posterior (in KL divergence). Therefore, one can view this GP approximation as a
>  combination of the prior and the posterior. Since the posterior is tunned to have the correct variance, while the prior has variance of O(1)
>  (it is not incluenced by the data), this results in a conservative (possibly overly large) approximation of the uncertainty in contrast to the common belief originating from the mean-field approximation.

---

> > ### Comment · Reviewer_KBMK · 2023-08-19
> >
> > The authors' reply has addressed my previous concerns, and I do see there are slightly more quantitative evaluations in the supplemental material. I've updated my score to 6.

---

> > > ### Author Response · Authors · 2023-08-21
> > >
> > > We would like to thank the Referee for the additional point.

---

### Official Review · Reviewer_VbCA · 2023-07-06

**Soundness:** 4 excellent
**Presentation:** 3 good
**Contribution:** 3 good
**Rating:** 5
**Confidence:** 2

**Summary:**

This paper analyses posterior contraction rates in Variational GP for mildly and severely ill-posed linear inverse problems. The authors prove that under certain conditions variational posterior achieves same contraction rates as the true posterior. Moreover, they show that given enough inducing points (in spectral domain) the above-mentioned condition holds.
As specific examples, the minimal number of inducing points needed for optimal contraction and contraction rate is derived for two mildly ill-posed problems (Volterra operator, Radon transformations) and one severely ill-posed problem (heat equation).

**Strengths:**

Analysis of posterior contraction rates improves our theoretical understanding of VGP for linear inverse problems.

The presentation is well structured.

The contribution is clear and well argued for.

**Weaknesses:**

The experimental part in the main manuscript is very limited, with most experiments taken out to the appendix.

I believe the paper is more suited for a journal format. It is very technical and requires a thorough review which is hardly possible in a conference format.

**Questions:**

Suggestions:
1. Increase readability of figure 1. The legend way too small.

**Limitations:**

The limitations are adequately addressed.

---

> ### Author Rebuttal · Authors · 2023-08-09
>
> Thank you for your interest in our paper. We acknowledge the brievity of the experimental part in the main paper due to the space limitations.
>  Please note that our work besides proposing a new methodological approach also provides theoretical guarantees in general settings and considers
>  three specific examples. This theoretical part is arguably the main contribution of our work. Therefore, we have decided to move most of the simulations
>  to the supplement in order to comply with the paper format. Nevertheless, we hope that the numerical results (although mostly moved to the supplement)
>  carry the intended message and illustrate our theoretical findings well.
>
> We agree with the reviewer that our paper is well-suited for a (good) Stats/ML journal. However, we have decided to submit our work to NeurIPS, because
>  this is one of the most prestigous machine learning conference receiving a lot of visibility from a broad audience with different backgrounds. We note
>  that variational Bayes and inverse problems are highly relevant for various community, including Machine learning, statistics and numerical analysis.
>   Therefore, in view of the multidisciplinary aspect of our work, we hope it would foster the links between these topics, and we deemed NeurIPS the best
>  venue to do so. As for the theoretical aspect of our work, we argue that theory is necessary when we seek to use ML techniques in a responsible way,
>  as it provides a justification for their use in practice and improves the explainability of the approach. Please also note that it is not uncommon
>  to publish theoretical and highly technical papers in NeurIPS and we made a big effort in rigorously deriving and presenting our proofs.
>
> Thank you for pointing out the readability issue with Figure 1, we have addressed it in our revision.

---

### Official Review · Reviewer_TgVX · 2023-07-25

**Soundness:** 3 good
**Presentation:** 3 good
**Contribution:** 3 good
**Rating:** 7
**Confidence:** 3

**Summary:**

This theoretical paper addresses the problem of inferring a function $f$ when

$y(x_i) = \[Af\](x_i) + \epsilon_i$ with $i=1, ...,n $

and $A$ is suitably defined linear operator with $\epsilon_i$ being i.i.d Gaussian noise. In words this is a linear inverse problem.

A (standard) Bayesian approach is adopted by putting a Gaussian process prior on $f$ with a carefully chosen covariance function.

Then an interdomain variational inducing point approach is adopted. All this so far is standard and pre-existing.

The contribution of the paper is to derive posterior contraction rates for this setup under certain assumptions and to apply it some specific examples with different choices of the the linear operator $A$. Specifically these are a volterra operator, a diffusion equation and a Radon transform.

**Strengths:**

This is an important and challenging topic. The standard regression case has been studied by Nieman et al. 2022 which corresponds to taking the $A$ to be the identity in this paper. The extension is certainly of interest since the Bayesian approach to inverse problems is a thriving and important area.

The required extension is treated well by the authors. The particular challenges of introducing an operator $A$ are clearly described and handled. The results derived require significant technical heavy lifting.

I was not able to go through the proof in detail but the results and approach look sensible and I suspect it to be correct.

The examples equations chosen were interesting and informative.

The numerical simulations, if brief, were nonetheless adequate in a paper of this type and compelling.

Limitations were communicated where they arose.

**Weaknesses:**

**The paper requires strong assumptions on the kernel of the Gaussian process.** To some extent these are forced by theory since a very mispecified covariance function would not contract. But, as the authors clearly concede, there is a more general set of covariance functions that would work than the ones considered.

**The two methods of choosing the inducing variables are somewhat idealized.** For a mildly ill-posed problem and the empirical spectral feature method the complexity is $O(N^2 M)$. Since $M$ the number of inducing points typically needs to be an increasing function of $N$ the number of data points then this is not much of an improvement on the exact inference which is $O(N^3)$ in practice and $O(N^{2.3})$ in theory. The authors do mention this. Alternatively the population method of choosing inducing variables uses $O(NM^2)$ but requires oracle knowledge of the eigensystem of $A^*A$. Admittedly the authors do show some interesting examples where they are able to do this. The required $M$ is more benign for the strongly ill-posed system and correspondingly the heat equation example given is quite compelling.

**The inducing inputs are not optimized** This is known to make a big difference in the more practical part of the literature particularly in higher input dimensions. It is also one of the things that distinguishes the Titisias variational approach from the Deterministic Training Conditional (DTC) approach using the nomenclature of this paper https://www.jmlr.org/papers/volume6/quinonero-candela05a/quinonero-candela05a.pdf.

**The literature review is not complete.**

There are two significant papers from Pinski et al. Specifically:

*Kullback--Leibler Approximation for Probability Measures on Infinite Dimensional Spaces. Pinski, Simpson, Stuart and Weber. 2015*

*Algorithms for Kullback--Leibler Approximation of Probability Measures in Infinite Dimensions. Pinski, Simpson, Stuart and Weber. 2015*

These papers are early, rigorous and are in a similar area to the submission. The method is distinct since they work principally with the inverse covariance operator. Importantly they also address the challenging non-linear case. The type of guarantee derived is also different since they do not derive contraction directly. Instead in the second they use their approximating for an MCMC proposal and gain asymptotic guarantees that way. Nonetheless these papers should be discussed.

A historical note. The work of *Burt et al. 2019* is excellent and highly impactful but perhaps a bit too much is attributed to it here. It provides *rates* of convergence to the posterior but there was some work on convergence before this. This is largely because convergence follows straightforwardly from the variational formulation. *Titsias 2009* (the longer version) Section 2.3 has some discussion of improvement in terms of KL. This was formulated in terms similar to those used in the present submission by *Matthews et al. AISTATS 2015 On Sparse variational methods and the Kullback-Leibler divergence between stochastic processes.* The AISTATS paper mentioned also references several earlier foundational papers and the present submission would benefit from mentioning them too. For instance the early work authored by *Opper*, *Seeger* and *Csato* amongst others. None of these existing papers removes the main contribution of the current work.

**Smaller points/ typos:**

For the mildly ill-posed case described in Definition 1 shouldn't the decay be polymonial rather than exponential?

On line 346 I believe the units need to be milliseconds for the description to make sense. So I think the typo is that "50.5 m" should be "50.5 ms".

**Questions:**

I have no questions other than the ones mentioned above.

**Limitations:**

There are no negative impacts emerging from this work that I can see.

Limitations are adequately discussed in the submission for the most part. The "weaknesses" section of this review has further discussion.

---

> ### Author Rebuttal · Authors · 2023-08-09
>
> We would like to thank the reviewer for carefully reading our paper and for the helpful comments.
>
> ## Assumptions on the GP kernel
>
> We thank the referee for this excellent comment. It is indeed standard in the literature to model covariance functions in accordance to the linear
>  operator $\mathcal{A}$. More general priors were also considered e.g. [1,39] but the covariance structure in all these cases were closely
>  related to the SVD bases of the operator $\mathcal{A}$. For example, wavelet priors can provide a feasible alternative in several examples [39].
> A general understanding, however, is still missing even for the true posterior, let alone its variational approximation. This poses an interesting,
>  but very challenging and technical research topic, going beyond the scope of our paper.
>
>
> ## Idealized choice of inducing variables
>
> We agree that in the mildly ill-posed case, the improvement from $\mathcal{O}(N^3)$ to $\mathcal{O}(NM^2)$ in the population and to $\mathcal{O}(N^2M)$ in
>  the empirical spectral features method (where the SVD basis of $\mathcal{A}$ is not available) is somewhat limited. However, we argue, that even this
>  relatively mild improvement allows us to analyse substantially bigger dataset in practice (and speeds up the computations as demonstrated in our numerical analysis).
> We completely agree with the referee, that in the case of severely ill-posed problems the improvement is substantial, allowing basically linear computational complexity.
>
> ## Inducing points are not optimised
>
> Yes, we completely agree, that inducing points methods are highly important in practice. However, their analysis is considerably more challenging
> and to the best of our knowledge no posterior contraction rates were derived even in case of the direct problem. Therefore we leave this important
> topic to future work and thank again for the suggestion.
>
>
> ## Literature review and typos
>
> We are particularly grateful to the reviewer for the two references from Pinsky et al. which are indeed closely related to our problem.
>  While their objectives are different from ours, we now cite them as suggested by the reviewer, to make the literature review more complete.
>  We have also included the reference to Seeger, Csato and Opper in our revised manuscript, thanks for pointing it out.
>
> Thank you for reporting these typos, we have corrected them in the revised manuscript.

---

> > ### Comment · Reviewer_TgVX · 2023-08-14
> > **Reply to authors**
> >
> > I thank the authors for their reply which I have read.

---

### Official Review · Reviewer_rih9 · 2023-07-26

**Soundness:** 3 good
**Presentation:** 3 good
**Contribution:** 3 good
**Rating:** 6
**Confidence:** 3

**Summary:**

The authors propose an inducing-point variational Bayes approximation for linear inverse problems. They study the frequentist properties of the arising variational posterior approximation. Specifically, they prove contraction rates for the variational posterior, both in the mildly and severely ill posed case. Moreover, they illustrate the practical relevance of their results with a number of synthetic data experiments.

**Strengths:**

The paper is well-written and its objectives and scope are clear. It provides an important contribution. As far as I am aware, frequentist properties of variational posteriors arising as inducing-point approximations in linear inverse problems have not been studied explicitly before. Such properties and contraction rates are useful in that they illustrate how the choice of the prior and the number of inducing points in the approximation impacts the statistical properties of the variational posterior. In that sense, they can provide useful guidance for the user.

**Weaknesses:**

I am not exactly sure if the comparison to the existing literature provided by the authors is substantial enough – I address this among my questions to the authors.

**Questions:**

In 2020, three papers were published in the Annals of Statistics providing a comprehensive and general study of the frequentist properties of variational posteriors. Two of those papers are cited by the authors as references [2] and [56]. Another one I would add to this list is “$\alpha$-VARIATIONAL INFERENCE WITH STATISTICAL GUARANTEES” by Yang, Pati, Bhattacharya. I wonder if the authors would explain whether the results of those papers could be applied to the variational GP approximations for linear inverse problems, as considered in the authors’ submitted work. If they could, then I would like the authors to give reasons why they decided to study this problem directly rather than applying the theory provided previously in those three papers. If they couldn’t, I would like the authors to briefly describe why.

I also have a few other questions and suggestions. In Definition 1 the authors say that a problem is mildly ill-posed if $\kappa_j$ has exponential decay. Do the authors mean polynomial decay? Also in Theorem 1, only at the end of the statement do the authors define what it means for a posterior (or a variational posterior) to contract at a certain rate. I would include a definition of what it means to contract at a certain rate before stating Theorem 1. Moreover, I’m not sure if the notation $S_{++}^m$ used at the bottom of page 4 and the notation $\Pi_u$ used at the top of page 5 are introduced anywhere in the paper before they are used.

**Limitations:**

A limitation of this work concerns its practical usability. For example, the authors’ results hold under the assumption that the model is well-specified. It is hard to expect that such an assumption will be satisfied in real-life applications and for real data sets. I believe that the authors should acknowledge this limitation and perhaps comment on possible extensions to instances of model misspecification.

---

> ### Author Rebuttal · Authors · 2023-08-09
>
> We would like to thank the reviewer for  his/her interest in our work and the suggestions.
>
>
> ## Literature review
>
> While the three papers on variational Bayes published in AoS in 2020 are indeed all cited in our
> article [2,54,56], they consider direct problems, while our focus is on ill-posed inverse problems where the standard approaches can not be applied directly.
> The challenge is that the operator \mathcal{A} does not possess a continuous inverse hence one natural way to connect the direct and inverse problems
> is through modolus of continuity [22]. The modulus of continuity measure quantitatively the continuity of the operator (on a subset) of the function space,
> see (26) for the precise definition in the Supplement. To have a meaningful bound and hence a good control on this quantity one has to consider appropriate
>  metrics both in the "inverse" $L_2(\mathcal{T},\mu)$ and "forward" $L_2(\mathcal{X},G)$ function spaces. A natural and (relatively) easy to handle choice is
>  the $L_2$-norm in both cases. A further difficulty of using standard testing based methods (on which the general approaches are typically built on) is that
>  they provide contraction rates in the direct, forward model with respect to the Hellinger distance or empirical $L_2$-norm, none of which are particularly
> appropriate to derive meaningful bounds for the modulus of continuity. Finally, we would like to note that for Gaussian processes (GP) a more specialised,
>  powerful machinary was developed in the literature based on GP concentration functions [52]. We advocate that stating the conditions in terms of the
> corresponding GP concentration inequality provides more insights than general approaches presented in the previous papers. This considerably simplifies
>  the treatment of the true posterior distribution. Then to connect the true posterior with its variational approximation one can use a simple formula based
>  on the duality of Kullback-Leibler divergence. In our case the most convenient formulation was derived in [40], but very similar techniques were used in
>  the earlier papers [2,54,56] as well.
>
>
> ## Typos and other minor comments:
>
> Thank you for pointing out this typo, it is indeed mildly ill-posed, we have corrected it. Based on your suggestion, we have now included the definition
>  of a posterior contraction rate before stating our main results. We have also updated the notation section with $S_{++}^m$ denoting the set of $m\times m$
>  positive-definite matrices and $\Pi_\mathbf{u}$ the induced prior distribution of the inducing variables $\mathbf{u}$.
>
> ## Misspecification
>
> We agree that studying the robustness of our methodology in case of misspecification is an interesting and important problem. In case of misspecified models
>  the posterior typically contracts around the Kullback-Leibler projection of the true distribution to the considered misspecified class. This has been well
>  established in parametric and semi-parametric frameworks, but in non-parametric models, let alone inverse problems (to the best of our knowledge) no formal general
>  result is available. Our focus here was on deriving the first contraction rate result for variational Bayes in case of ill-posed inverse problems,
>  hence we leave this extension to future research.

---

> > ### Comment · Reviewer_rih9 · 2023-08-16
> > **Reply to authors**
> >
> > The authors have addressed all my comments and questions. I will keep my evaluation as it is. I must say, however, that having read the comment of reviewer VbCA I can't help but agree with them that such highly technical papers might be better suited for publication in Statistics/ML journals. It is difficult to check complicated mathematical arguments carefully within the very tight schedule of the NeurIPS reviewing process. The authors' results look correct but I admit I did not manage to check their proofs carefully and from what I see nor did some of the other reviewers.

---

### Author Rebuttal · Authors · 2023-08-09

# General rebuttal:

We would like to thank the reviewers for carefully reading our manuscript, for acknowledging the importance of the topic
 ("important and challenging topic" - TgVX) and that the examples are " interesting and informative" (TgVX).
We are happy to hear that they found the paper "clear and well argued" (VbCA), "well-written" (KBMK, rih9), and that it
 "provides an important contribution" (rih9). We would also like to thank the reviewers for their constratice comments
 and suggestions which lead to a further improved version of the manuscript.

We highlight here the main changes. Below, we address the comments of each referee point-by-point and provide a more detailed answer to their questions.
- We have corrected the typos and provided the few missing definitions/concepts. We also discuss the strength and limitations of our results in more details.
- We have provided an extended description of the pracical application of the considered examples following the suggestion of KBMK.
- Following the suggestion of KBMK we have compared numerically the empirical and theoretical "phase-transition curves".
We have conducted a numerical experiment for the Voltera operator, please find the figure attached. We computed the (logarithm
of the) fraction of the mean integrated squared error (MISE) corresponding to the true and variational posterior means (we used 20 repetitions to
 empirically approximate these quantities). We have considered n ranging from 100 to 10000 and m from 1 to 17. We have also plotted the phase transition
 curve (white line) coming from our theoretical analysis (see the PDF file attached). One can note that the theoretical curve closely resembles the curve where the phase transition occurs
in the empirical study. We believe that this provides a further evidence that our theoretical results indeed turn out to be a useful practical guide.

---

### Decision · Program_Chairs · 2023-09-21

**Decision:**

Accept (poster)

**Comment:**

The reviewers agrees that the paper provides new theoretical insights into the tuning of sparse variational Gaussian process models for solving linear inverse problems, which arise frequently in scientific applications.